# Online dynamical downscaling of temperature and precipitation within the *i*LOVECLIM model (version 1.1)

Aurélien Quiquet[1], Didier M. Roche[1,2], Christophe Dumas[1], and Didier Paillard[1]

[1]Laboratoire des Sciences du Climat et de l'Environnement, LSCE/IPSL, CEA-CNRS-UVSQ, Université Paris-Saclay, F-91191 Gif-sur-Yvette, France
[2]Earth and Climate Cluster, Faculty of Earth and Life Sciences, Vrije Universiteit Amsterdam, Amsterdam, the Netherlands

*Correspondence to:* A. Quiquet (aurelien.quiquet@lsce.ipsl.fr)

**Abstract.**

This paper presents the inclusion of an online dynamical downscaling of temperature and precipitation within the model of intermediate complexity *i*LOVECLIM v1.1. We describe the followed methodology to generate temperature and precipitation fields on a 40 km x 40 km Cartesian grid of the Northern Hemisphere from the T21 native atmospheric model grid. Our scheme is non grid-specific and conserves energy and moisture in the same way as the original climate model. We show that we are able to generate a high resolution field which presents a spatial variability in better agreement with the observations compared to the standard model. Whilst the large-scale model biases are not corrected, for selected model parameters, the downscaling can induce a better overall performance compared to the standard version on both the high-resolution grid and on the native grid. Foreseen applications of this new model feature includes ice sheet model coupling and high-resolution land surface model.

## 1 Introduction

In recent decades, the Earth has undergone a sustained global warming due to a rapid rise in greenhouse gases, a rise unprecedented over the last million years (Luthi et al., 2008; Wolff, 2011). Some components of the Earth system, such as the oceanic and terrestrial carbon cycles or the continental ice sheets, present feedbacks acting over long timescales, i.e. multi-millenial, and are suspected to play an important role for the climate in the future (Archer and Brovkin, 2008). Earth models of intermediate complexity (EMICs) are powerful tools to investigate the long-term transient response of the climate system (Claussen et al., 2002). The advantage of EMICs is to include most of the major climatic components in a unified and coupled framework whilst being computationally inexpensive compared to more comprehensive general circulation models (GCMs) because of a simplified physics and a coarser resolution. As such, they can be used to perform numerous simulations to assess model sensitivities (e.g. Loutre et al., 2011) or multi-millenial integrations to study slower feedbacks responses (e.g. Calov et al., 2005). EMICs were initially developed as computationally cheap alternatives to general circulation model especially in the context of studying the role of orbital and carbon dioxide forcing and feedback within the context of glacial-interglacial cycles (e.g. Weaver et al., 1998; Berger et al., 1998; Ganopolski et al., 1998). With the addition of interactive ice sheets, EMICs became capable of studying ice sheet dynamics in term of retreat, advance and stability as a key component of the climate system (e.g. Calov et al., 2002; Huybrechts et al., 2002; Charbit et al., 2005). Also, some EMICs include an interactive carbon cycle which

allows the investigation of the mechanisms behind the atmospheric carbon dioxide fluctuations during the Quaternary (e.g. Brovkin et al., 2007; Ridgwell and Hargreaves, 2007; Bouttes et al., 2011). However, with increases in computing facilities, EMICs are generally becoming more comprehensive than they have ever been. From zonally averaged atmosphere or ocean (e.g. Gallée et al., 1992; Petoukhov et al., 2000), they now often include a three dimensional ocean (e.g. Edwards and Marsh, 2005; Weaver et al., 2001). Yet, the atmospheric component has remained a simplified component in EMICs even though they may be sometimes three dimensional but with only a limited number of vertical levels and slightly simplified base equations (e.g. Goosse et al., 2010).

However, the relative simplicity and coarse resolution of such climate models result in an approximative representation of land surface climatic variables that show a high spatial variability. Precipitation is an example of such a variable, being a key component of the climate system and nonetheless generally poorly represented in atmospheric models. In particular, EMICs are unable by design to reproduce correctly the meso-scale atmospheric processes induced by relatively fine-scale topographic features such as mountain ranges. This has important consequences for the sub-components of the climate system that depend on the atmospheric water cycle such as surface hydrology, vegetation or water isotopes. Higher resolution is thus imperative for components whose large-scale physical behavior is highly dependent upon processes occurring at small spatial scales. The limitations induced by coarse resolution has led to it becoming a recurrent issue in climate-hydrology studies at basin scale (e.g. Vetter et al., 2015) as well as in ice sheet - climate coupling studies (e.g. Charbit et al., 2005; Fyke et al., 2011).

In particular, ice sheet models require a higher resolution to account for the narrow ablation zones at the ice sheet margins (Ettema et al., 2009). To account for it, ice sheet – climate coupled models have often preferred to use their own anomalies regridded on top of a reference climate to force the ice sheet model (e.g. Vizcaíno et al., 2008; Goelzer et al., 2016). The anomalies are then linearly interpolated and added to well-constrained and high-resolution present-day climate fields. Such a strategy implicitly assumes that the model biases remain unchanged through time and are independent from the imposed external forcings, and also remain unchanged as ice sheet geometry changes significantly. Alternatively, another strategy is to use absolute fields, but downscaled to the needed resolution. The complexity of such downscaling approaches ranges from simple bi-linear interpolations (e.g. Vizcaíno et al., 2010; Gregory et al., 2012) to more physically based approaches. To achieve temperature downscaling, Charbit et al. (2005) duplicate the energy budget calculation on 15 artificial levels in order to retrieve surface temperature on a vertically extended grid. Whereas Fyke et al. (2011) follow a similar strategy but in addition derived the precipitation on the vertical extended grid. Alternatively, Robinson et al. (2010) embed a simplified regional energy-moisture balance model in an EMIC in order to assess sub-grid processes unresolved by their native atmospheric model. Although statistical downscaling has been applied to EMIC outputs (Vrac et al., 2007; Levavasseur et al., 2011), these techniques were not used to couple the various different components of models.

Here, we present the inclusion of a relatively unexpensive online and conservative dynamical downscaling of temperature and precipitation in the *i*LOVECLIM coupled climate model (version 1.1). The downscaling is done from the native T21 grid

($\simeq 5.625°$ spatial resolution) towards a cartesian 40 km x 40 km grid of the Northern Hemisphere. The chosen high resolution grid arises from the ice sheet model grid embedded in *i*LOVECLIM (Roche et al., 2014). The methodology chosen for the downscaling procedure is to first replicate the original model physics on artificial surfaces of a vertically extended grid. Then from the vertically extended grid, we compute the precipitation explicitly taken into account the sub-grid orography following

the original model physics. Computed on each atmospheric timestep, the downscaling accounts for the feedback of sub-grid precipitation upon the large scale energy and water budget. Whilst the energy repartition between the turbulent fluxes is modified, the conservation is ensured however in the same way as in ECBilt, where the heat flux towards land and ocean is computed as the imbalance between the incoming (both shortwave and longwave) and the outgoing radiation (longwave only) as well as the turbulent (latent and sensible) heat fluxes.The conservation of energy and water is particularly important for multi-millenia

simulations. As the downscaling methodology is not grid-specific and it can be applied in the future to any grid having a higher resolution than the native T21 grid. In particular, downscaling over a specific region (e.g. Europe or the Andes) is possible with our implementation. Foreseen future applications include ice-sheet surface mass balance computation and land surface modelling (hydrology, permafrost, vegetation dynamics and land carbon) at continental scale and high resolution.

In Sec. 2 we describe the implementation of the dynamical downscaling of temperature and precipitation in the atmospheric component of the *i*LOVECLIM model. In Sec. 3 we discuss the performance of both the standard and downscaled temperature and precipitation fields in representing present-day climatological fields. We list concluding remarks and perspectives in Sec. 4.

## 2   Methodology

### 2.1   the *i*LOVECLIM model

*i*LOVECLIM (here in version 1.1) is a code fork of the LOVECLIM 1.2 model, extensively described in Goosse et al. (2010). Whilst the physics in the atmosphere, ocean and land surface has remained mostly unchanged, the major bifurcations from Goosse et al. (2010) consist in the addition of a water oxygen isotope cycle (Roche, 2013; Roche and Caley, 2013), an oceanic carbon model (Bouttes et al., 2015), an alternative ice sheet model (Roche et al., 2014), the reimplementation of the initial iceberg model (Bügelmayer et al., 2015), and a permafrost model (Kitover et al., 2015). The LOVECLIM family models contain

a free surface ocean general circulation model with an approximately three degrees spatial resolution resolution and 20 vertical layers. It is coupled to a thermo-dynamical sea ice model operating on the same spatial grid. The atmospheric component of main concern here, ECBilt, is a quasi-geostrophic model, solved on a T21 spectral grid. For a complete description of ECBilt, the reader is referred to Haarsma et al. (1997) and Opsteegh et al. (1998) and references therein. The dynamics, i.e. the resolution of the potential vorticity equation, is computed for three vertical levels: 800 hPa, 500 hPa and 200 hPa. The equations for

temperature and vertical motion are computed on two intermediate levels at 650 hPa and 350 hPa. A schematic representation of the vertical structure of the atmosphere in ECBilt is shown in Fig. 1.

The main idea of the downscaling procedure is to replicate the processes governing precipitation formation and surface temperature computation on a refined vertical extended grid in order to assess these variables at any altitude for any given sub-grid.

## 2.2 Vertical profiles of temperature and moisture

The first steps of the downscaling is to recompute temperature and moisture variables on artificial surfaces of a vertically extended grid of the atmosphere.This grid consists in 11 vertical levels at 10, 250, 500, 750, 1000, 1250, 1500, 2000, 3000, 4000 and 5000 m. In the following, we present the equations already described in Haarsma et al. (1997), which are needed for the vertically extended grid.

### 2.2.1 Temperature profile

In ECBilt, due to the lack of a proper representation of the atmospheric boundary layer, an idealised vertical profile is used to compute heat, moisture and momentum fluxes at the Earth surface. Above 200 hPa, the atmosphere is assumed to be isothermal. Assuming hydrostatic equilibrium and using the ideal gas law, the temperature varies linearly with the logarithm of pressure. For this reason, from the 650 hPa and 350 Pa intermediate levels, we compute this linear temperature profile in the logarithm of pressure from 200 hPa to the surface.

Thus, for any pressure level $p$, the temperature is:

$$T(p) = T_{650} + \gamma ln\left(\frac{p}{p_{650}}\right) \tag{1}$$

With $\gamma$ the atmospheric temperature lapse rate as:

$$\gamma = \frac{T_{350} - T_{650}}{ln\left(p_{350}/p_{650}\right)} \tag{2}$$

In Haarsma et al. (1997), the near-surface air temperature of an atmospheric grid cell, $\bar{T}_*$ , is computed from $T_{500}$, using Eq. 1
to eliminate the pressure variable in the hydrostatic equilibrium equation:

$$\bar{T}_* = \sqrt{T_{500}^2 - \frac{2\gamma g}{R}\left(\bar{z_h} - z_{500}\right)} \tag{3}$$

With $\bar{z_h}$ is the grid-cell surface height and $z_{500}$ the height of the 500 hPa levels (prescribed homogeneously at 5500 m).

This equation is used to assess the near-surface air temperature for the 11 artificial surfaces using explicitly their altitude,
$z_h\left(l = 1, 11\right)$, instead of the actual surface height of the grid cell:

$$T_*\left(l = 1, 11\right) = \sqrt{T_{500}^2 - \frac{2\gamma g}{R}\left(f_s z_h\left(l\right) - z_{500}\right)} \tag{4}$$

The vertical lapse rate in temperature computed in the model in Eq. 2 is representative of the free-atmosphere temperature variations. Since the along-slope lapse rate is generally smaller than the free-atmosphere lapse rate (e.g. Marshall et al., 2007; Gardner et al., 2009; Minder et al., 2010), its use lead to an overestimation of the temperature changes with elevation. In order

to artificially reduce the value of the vertical lapse rate in the model, we apply a global tunable correcting factor, $f_s$ in Eq. 4 (typically ranging from 0.5 to 1.), to the orography on the vertically extended grid.

From this near-surface air temperature for the artificial surfaces, we derive several surface energy balance terms (downward longwave radiation, latent and sensible heat flux) in the same way as Haarsma et al. (1997). Surface temperatures at the artificial surfaces $T_s$ $(l = 1, 11)$ are computed iteratively from the energy balance, assuming a zero heat capacity of the surface. We assume no change in surface types, and consequently albedo, between the different artificial layers. Because the latent heat flux depends on the evaporation, we also need to assess the specific humidity at the 11 artificial surface levels.

### 2.2.2 Moisture profile

In the idealised ECBilt representation of the atmosphere, only the lower part of the atmosphere (i.e. below 500 hPa) contains water. A single equation is used to compute the evolution of total precipitable water $\bar{q}_a$ from advection, precipitation and evaporation. In our version of the model, precipitation occurs when the total amount of precipitable water is greater than a fraction ($\alpha_q = 90\%$) of the vertically integrated saturation specific humidity $q_{max}$. For each artificial level, the expression of $q_{max}$ $(l = 1, 11)$ is computed as in Haarsma et al. (1997) as the vertical integral of the saturation specific humidity in the pressure coordinate:

$$q_{max}\left(l = 1, 11\right) = \frac{1}{\rho_w g} \int\limits_{p_0(l)}^{500hPa} q_s\left(T, p\right) dp \tag{5}$$

Where $\rho_w$ is the water density and $g$ is the gravitational acceleration. The surface pressure $p_0$ $(l = 1, 11)$ is computed rearranging Eq. 1 in term of pressure and using Eq. 2:

$$p_0\left(l = 1, 11\right) = p_{650} \, exp\left(\frac{T_*\left(l\right) - T_{650}}{\gamma}\right) \tag{6}$$

The saturation specific humidity at a given level, $q_s\left(T, p\right)$, is given by a Clausius-Clapeyron expression of the saturation vapour pressure. The vertical profile of specific humidity is retrieved assuming a constant relative humidity for the whole atmospheric column below 500 hPa.

### 2.3 Sub-grid precipitation and coarse grid upscaling

From the climatic variables computed on the artificial surfaces on the vertically extended grid, we can compute the precipitation and temperature at the sub-grid orography.

### 2.3.1 From the vertically extended grid to the sub-grid

For a given native coarse-grid point at a given surface height $\bar{z}_h$, we have a certain numbers of sub-grid points $k$ of different surface heights $z_h$ $(k = 1, k_{max})$. The surface elevation of the native grid comprises the area-weighted average of all k sub-grid

points:

$$\bar{z}_h = \frac{\sum_{k=1}^{k_{max}} (z_h(k)s_a(k))}{\sum_{k=1}^{k_{max}} s_a(k)} \tag{7}$$

Where $s_a(k)$ is the area of the sub-grid cell.

In order to compute the heat and moisture budget on a sub-grid point $k$, we linearly interpolate a needed surface variable $\phi$ from the bounding vertical levels $l$ and $l+1$. Thus, from the variables computed on the vertically extended grid, we recompute on the sub-grid: the near-surface air temperature $T_*$, the surface temperature $T_s$ and integrated saturation specific humidity $q_{max}$.

Winds are not downscaled in our approach. In the real world, orographic precipitation mostly occurs on wind-faced slopes

whilst the other side is generally much drier. On the native grid of ECBilt, winds transport humidity and thus affect precipitation amounts. For our downscaling approach, because winds are not downscaled, in order to mimic the enhancement of precipitation on wind-faced slopes, we sort the sub-grid points by elevation for a given coarse grid point so that the lowlands before the mountain ranges are treated before the higher altitudes. The lowest grid point is initialized with the coarse-grid value: $q_a(k=1)=q_a$ . As we compute precipitation for a sorted sub-grid point, we remove available precipitable water from

the amount of total precipitable water of the previous grid point. In doing so, we assume that the mountain edges (lowest elevations) are the first affected by moisture influx. However, in our approach two points at the same altitude will have the same amount of precipitation, independantly from the wind direction. The model is thus intrinsically unable to reproduce high precipitation on windward slopes and conversely low precipitation on leeward slopes. A foreseen model development will be to sort the sub-grid points depending on wind direction.

### 2.3.2 Stratiform precipitation

Two processes are responsible for stratiform precipitation in ECBilt. First, since the upper atmospheric layer (above 500 hPa) is assumed to be dry, any vertical moisture export through the 500 hPa level is converted into precipitation. The amount of this export is calculated from the moisture availability at 500 hPa, which depends of the local surface topography. For this reason,

we expand the computation of moisture export on the vertically extended grid. Following a similar expression as in Haarsma et al. (1997), in case of a negative vertical velocity at 500 hPa, $\omega$, the amount of precipitation is computed as the export of moisture outside the 500 hPa level:

$$p_{dyn,ve}(l=1,11) = -\omega q_*(l)/\rho_w g \tag{8}$$

where $q_*$ the precipitable water given by:

$q_*(l=1,11) = r(l)q_s(p=500\ hPa)$                                          (9)

with $r$ the relative humidity. For a given grid point, the relative humidity shows a constant vertical profile. However, its value depends on the local topography since its computation is derived from the vertically integrated saturated specific humidity

(Eq. 5):

$$r\left(l=1,11\right)=q_a/q_{max}(l) \tag{10}$$

From the stratiform precipitation on the vertically extended grid, $p_{dyn,ve}\left(l=1,11\right)$, we compute the corresponding sub-grid precipitation, $p_{dyn,ve}\left(k=1,k_{max}\right)$, with a linear interpolation from the bounding vertical levels.

Another contribution to stratiform precipitation is due to moisture excess. In the version of ECBilt included in iLOVECLIM v1.1, stratiform precipitation occurs when the total amount of precipitable water, is greater than $\alpha_q=90\%$ of the vertically integrated saturation specific humidity. On the sub-grid points a similar condition is checked, based on the local total amount of precipitable water, $q_a\left(k=1,k_{max}\right)$, and the local vertically integrated saturation specific humidity $q_{max}\left(k=1,k_{max}\right)$. In

the original version of ECBilt, the value for $\alpha_q$ has been tuned to reproduce the global scale precipitation pattern. Because of the higher spatial variability in topography, the downscaling induces a change in the precipitation pattern. There is no reason why this tuned $\alpha_q$ should be kept unchanged from the original model. In addition, because of the strong non-linearity of the precipitation to elevation, we add the possibility to modify the value of $\alpha_q$ depending on the local elevation $z_h\left(l=1,k_{max}\right)$:

$$\alpha_q\left(k=1,k_{max}\right)=min\left(\alpha_q^{min}+\left(1-\alpha_q^{min}\right)\frac{z_h(k)}{z_q},1\right) \tag{11}$$

where $\alpha_q^{min}$ is the value for a point at sea level and $z_q$ is the altitude above which the precipitation occurs only if the total precipitable water reaches 100% saturation. As in Haarsma et al. (1997), stratiform precipitation due to moisture excess is expressed as:

$$p_{dyn,mc}\left(k=1,k_{max}\right)=\frac{q_a-\alpha_q(k)q_{max}(k)}{C_{lh}(k)*dt} \tag{12}$$

With $dt$ the atmospheric model timestep (4 hours) and $C_{lh}$ a corrective term to account for latent heat release in the atmosphere

associated with the precipitation:

$$C_{lh}\left(k=1,k_{max}\right)=1.+\frac{r(k)\rho_w L_c g}{c_p\Delta p_l}\left(\frac{dq_{max}}{dT_{650}}\right)(k) \tag{13}$$

With $L_c$ the latent heat of condensation, $c_p$ the specific heat capacity and $\Delta p_l$ the lower layer depth (500 hPa). $\frac{dq_{max}}{dT_{350}}$ is obtained from tabulated values of Eq. 5.

For the two contributions of stratiform precipitation, the near-surface air temperature of the sub-grid, $T_*\left(k=1,k_{max}\right)$ , is used to determine snow and rain partition with an abrupt transition at $0\,^{\circ}C$. Similarly to what is done for coarse grid precipitation in the standard version of ECBilt (Haarsma et al., 1997; Opsteegh et al., 1998), the sub-grid stratiform precipitations, either snow and rain, are associated with a local release of heat at 350 hPa, modifying $T_{350}\left(k=1,k_{max}\right)$.

### 2.3.3 Convective precipitation

Convective precipitation is assumed to be an adjustment term to reach stability in the atmospheric column. They represent roughly 10% of the total precipitation in the model. We compute convective precipitation after the stratiform precipitation.

If the moisture availability $q_a\left(k=1,k_{max}\right)$ is still greater than $\alpha_q\left(k\right)q_{max}\left(k\right)$ then the amount of convective precipitation, $p_{conv}\left(k=1,k_{max}\right)$, is computed with the same formulation as in Eq. 12. As for the stratiform precipitation, the convective precipitation is associated with a local heat release affecting the temperature at 350 hPa, $T_{350}\left(k=1,k_{max}\right)$. After this convective precipitation, we assess stability comparing the moist adiabatic lapse rate to the local potential temperature at 500 hPa,

$\theta\left(k=1,k_{max}\right)$, computed from the potential temperatures at 350 hPa and 650 hPa. The stability is assessed for each individual sub-grid points. If the stability is not reached, we allow a new convective precipitation term computed from $q_a\left(k=1,k_{max}\right)$. The heat release in the upper atmosphere at each precipitation event tends to increase stability. This is an iterative process and we only go to the next sub-grid point when we reach stability locally.

### 2.3.4   Upscaling to the coarse grid

Following the stratiform and convective iterations on the sub-grid, moisture and energy on the native grid have to be updated. On the one hand, the initial coarse-grid moisture is simply reduced by the sum of sub-grid total precipitations, hence readily conserving water. On the other hand, the temperatures at 350 hPa and 650 hPa are recomputed as the mean of the sub-grid temperatures at these levels.

## 3   Application and validation

### 3.1   Sub-grid of the Northern Hemisphere

As an example application, we use a sub-grid domain covering a large part of the Northern Hemisphere (hereafter NH40, Fig. 2). The sub-grid topography comes from ETOPO1 (Amante and Eakins, 2009), projected with a Lambert equal-area projection onto a squared 40 km x 40 km Cartesian grid. The grid contains 241x241 points with more than half of the domain being continental areas. This grid was chosen because it corresponds to the ice sheet model grid embedded in *i*LOVECLIM.

The T21 topography depicted in Fig. 2 corresponds to the NH40 topography aggregated to the native model resolution. This is the topography seen by the model when the downscaling is not performed.

### 3.1.1   Experimental design

For model evaluation, we define a control simulation (hereafter CTRL) as a 100 years of *i*LOVECLIM integration under

constant pre-industrial external forcing, branched to the standard long-term equilibrated pre-industrial restart. With the same experimental design, we define a series of downscaling experiments (hereafter DOWN) in which we compute the temperature and precipitation on the NH40 grid. For these experiments, we test the importance of three selected parameters: the elevation from which 100% saturation is needed to initiate precipitation $z_q$ in Eq. 11 (2000 and 3500 m), the minimum fraction of saturation to initiate precipitation $\alpha_q^{min}$ in Eq. 11 (0.7, 0.75, 0.8, 0.85, 0.9) and the lapse rate scaling factor $f_s$ in Eq. 4 (0.6,

0.7, 0.8, 0.9 and 1.). We explore the whole matrix of runs, which corresponds to 50 model realisations.

### 3.2 Model evaluation

For model evaluation, we compare the modelled annual mean climatic fields, namely surface temperature and precipitation rate, to observation-derived dataset. For this, we use a 1970-1999 climatological mean of annual surface temperature of ERA-interim reanalysis (Dee et al., 2011) and the long-term mean climatology of annual precipitation of CRU CL-v2 (New et al., 2002). We use ERA-interim on the $0.125°$x$0.125°$resolution for the whole Northern Hemisphere, whilst CRU CL-v2 covers the whole continental areas on a 10 min grid. We use bilinear interpolation to generate this data on the NH40 grid. For diagnostic purposes we also aggregate this data on the T21 grid with the same grid correspondance already used in Roche et al. (2014).

### 3.2.1 Surface temperature

The annual mean surface temperature for ERA-interim and model outputs on the NH40 and T21 grids is presented in Fig. 3. On the one hand, the general pattern, i.e. the strong latitudinal cooling, is generally well represented in the CTRL experiment. Whilst the model reproduces the cold temperatures in Siberia, it is elsewhere generally largely too warm, in particular over North America, Greenland and Western Europe. The temperature anomaly induced by local topography in the CTRL experiment is also largely underestimated. On the other hand, at the continental scale, our downscaling procedure does not imply important changes in surface temperature relative to the CTRL experiment. This suggests that the downscaling has only a minor impact on atmospheric circulation. However, the downscaling induces important local temperature changes, particularly visible on the NH40 grid. At this resolution, the temperature is reduced according to the local elevation. In many locations, the native grid is still visible on the NH40 model results. The imprint of the native grid remains because the primary effect of the downscaling is to physically compute the distribution of the climatic variables linked to temperature and precipitation according to the sub-grid topography for a given coarse grid information. By design, this generates discontinuities when moving from two neighbouring cells. Only air advection, which tends to be larger along parallels than meridians, reduces the imprint of the coarse grid.

In Fig. 4, we present the annual mean surface temperature for a selection of downscaling experiments accross selected transects: West to East for Europe and North America and South to North for Greenland (dashed purple lines in Fig. 3). ERA-interim temperature shows a strong dependency to elevation. This depency is remarkably well reproduced for the European transect. However, the warm model bias is only reduced for elevated areas, with only a very limited change at lower elevation. This is because our downscaling methodology strongly relies on topography and is thus not designed to correct broader region model biases that are unrelated to topographic forcing. For the other transects, even if the horizontal temperature gradients are generally better reproduced with the downscaling, the large model bias in the original model induces large errors, only slightly corrected by the downscaling.

To assess general model performance, we present in Fig. 5 a normalised Taylor diagram computed from ERA-interim and several model outputs. In this figure, we present one selected downscaling experiment (with parameter values: $z_q = 2000m$,

$\alpha_q^{min} = 0.8$, $f_s = 0.6$), as the sensitivity of the Taylor diagram to model parameters is very limited. Overall, the model generally shows very good skills in reproducing annual mean surface temperatures, for both the CTRL and DOWN experiments (filled circles). In particular, the model presents a good spatial correlation (greater than 0.9) with a standard deviation only generally slightly overestimated. Because the downscaling does not directly affect the climatic fields at low elevation, we also present

in Fig. 5 a normalised Taylor diagram computed from the montainous grid points (elevation greater than 800 m – triangles) only. With this, we can conclude that whilst the downscaling increases the agreement with ERA-interim for mountainous grid points, its impact for the whole grid is relatively limited. Interestingly, with and without the downscaling, the performance of the model is better when the lowlands are discarded. As the major model biases are located within these areas (e.g. more than 10 degrees around Hudson Bay). Finally, on the native model grid (outlined-only circles), the downscaling does not impact

significantly the model performance.

### 3.2.2 Precipitation

The annual mean precipitation rate for both the CRU CL-v2 and the model are shown in Fig. 6. The model reproduces some of the major large scale structures: East to West decrease in precipitation from the Eastern coast of North America, wet Rocky mountains and relatively wet Western Europe. The model however presents important biases in some places. In particular,

Eastern Siberia, the Southern part of the Rocky mountains and Eastern North America are wetter than the CRU CL-v2 dataset. Conversely the model is too dry in Eastern Europe or central North America regions. Generally, the CTRL simulation fails at reproducing the precipitation maximas over topographic features. The downscaling produces much more spatial variability, with its main effect being to increase the precipitation over elevated areas. Therefore, we are able to mimic the precipitation pattern in Western Europe with precipitation maximas over the Alps, the Scandinavian moutains or the British Highlands

(Fig. 7). However, the corresponding precipitation maximas in the observations do not necessarily perfectly coincide with the simulated ones: in the observations, the windward coasts experience generally more precipitation than the interior grid cells. This is particularly visible in the very narrow band (less than 200 km) of extremely high precipitation rate on the Western part of North America and along the Norwegian coast in the CRU CL-v2 dataset. Because, we do not take into account the winds in our approach, the main effect of the downscaling is to redistribute the precipitation according to the local topography within

a native T21 grid cell. In order to better resolve the fine scale structures, a redistribution of precipitation according to the wind direction could be in future versions a significant improvement. Over Greenland, the pattern obtained with the downscaling is much better than in the standard version with an increased South to North precipitation decrease (Fig. 8). Although the Northern part of Greenland is still wetter than the observations, it is drier than in the standard version of the model. Over the Rocky mountains, the downscaling reproduces some of the local features (Columbia mountains high precipitation), however,

the intrinsic model biases are generally not corrected. Where the model tends to be too wet (Eastern Siberia, Alaska or Southern Rocky mountains) the downscaling experiments are generally also too wet. This is particularly true where the topography is pronounced (Southern Rocky mountains). This means that the model large scale structures are generally stable and are only slightly impacted by the downscaling. In fact, the first order effect of the downscaling is to redistribute the precipitation according to the topography in a physically consistent way. In fact, there is only a relatively small change in the total amount of

precipitation when using the downscaling as the 30N to 90N averaged precipitation in the experiments presented in Fig. 6 is only decreased by 2% in this case.

In Fig. 9, we present the annual mean precipitation rate accross selected transects, revealing for all the selected transects, but in particular in Europe, the CTRL experiment presents too smooth variations in the rate of precipitation. The different downscaling versions simulate much more variability, coinciding with topography variations. In Europe the fit with observations is relatively good, one likely explanation could be the relatively small bias in the CTRL experiment within this region. However, an East- West divide exists in North America in which downscaling improves the precipitation in the East, but leads to an increase in the wet bias present in CTRL in the West. For Greenland, the CTRL simulations produce a precipitation maxima at the summit of the ice sheet which corresponds to the precipitation minima in CRU CL-v2. Conversely, the Western flank of the ice sheet for this transect is too dry in the CTRL experiment. Downscaling however considerably increases the precipitation along the western margin and produces a meridional precipitation gradient that is in better agreement with the observations. Through specific parameter combinations, it is possible to reduce the wet bias in the central part of the ice sheet, however the model is still largely too wet over central Greenland, perhaps due to dynamical features not captured by the T21 grid: the coarse resolution facilitates the advection of warm and moist air at the summit of the ice sheet.

A quantitative analysis of model performance is shown on Fig. 10 in which we present normalised Taylor diagrams for the CTRL and a selection of DOWN experiments against CRU CL-v2. On the NH40 grid (filled circles), most of the downscaling experiments improve model performance on one specific metric but not necessarily the others. In particular, a lower value for $\alpha_q^{min}$ tends to reduce the RMSE and to increase the spatial correlation, whilst the standard deviation is reduced. A lower value for $f_s$ also reduces the RMSE and the standard deviation but has almost no impact on the correlation. The parameter $z_q$ has a similar effect, but smaller in amplitude, than $f_s$ in the range tested. The real benefit of the downscaling is the better representation of precipitation for mountainous grid cells (elevation greater than 800 m – filled triangles). In this case, all the downscaling experiments present a better agreement with CRU CL-v2. The spatial correlation is in particular generally greatly improved (from about 0.25 to more than 0.4). On the original model resolution (outlined-only symbols), some selected downscaling experiments present an overall improvement. Generally, the downscaling has a non negligible impact on the precipitation fields on the T21 grid. For multi-millenia integrations, these changes on the hydrological cycle can have important feedbacks on the simulated climate. Whilst it is potentially prudent that a new tuning of the model parameters should be performed however in the meantime in order to avoid this, and for further applications, the parameter combination $z_q = 2000m$, $\alpha_q^{min} = 0.8$ and $f_s = 0.6$ is preferred because they produce an overall improvement of all metrics on the NH40 grid whilst they have a very minor changes from the CTRL experiment on the T21 grid.

The downscaling performance with respect to CRU CL-v2 is also shown in Fig. 11 in which we present quantitative metrics (spatial correlation, standard deviation and root mean square error) as a function of parameter values. The parameters that have the strongest influence on the simulated precipitation are $f_s$ and $\alpha_q^{min}$. A lower value for these parameters tend to produce

higher spatial correlation, lower standard deviation and lower root mean square error. However, for $z_q = 2000m$, low values for the two other parameters can lead to an underestimation of the standard deviation. The standard deviation and the root mean square error have a similar response to a change in parameters, whilst the spatial correlation is mostly sensitive to the $\alpha_q^{min}$ parameter, with higher correlation for lower value of this parameter.

## 4   Summary and perspectives

We have presented the inclusion of a dynamical downscaling of temperature and precipitation on a 40 km by 40 km grid of the Northern Hemisphere into a T21 resolution atmospheric model of intermediate complexity. The methodology chosen for the downscaling procedure replicates the relevant parts of the model physics needed for the temperature and precipitation on

the high resolution grid. An upscaling is performed from the high resolution precipitation and temperature, which takes into account the climatic feedback of sub-grid precipitation on the native grid climate. The scheme is conservative and, as such, is suitable for long-term integration.

We tested various parameters related to the temperature and precipitation at high resolution. The temperature is only locally

impacted by the downscaling with a cooling over mountainous areas. For the precipitation, we have shown that we are able to generate a field at high resolution which presents a better agreement with observations compared to the native coarse resolution atmosphere for mountainous region.The downscaling drastically increases spatial variability compared to the standard version of the model. Downscaling is however, unable to correct for large scale model biases, including biases in atmospheric circulation and model simplification, such that model performance is best when the biases in the standard version are low. In

particular, the model presents currently only one moist layer and has no explicit representation of clouds. Further development could include an iterative scheme for clouds and that these clouds could relate to precipitation. Such a development could be tested in the high resolution grid with a specific calibration of convective clouds based on topography. Another model limitation is the lack of diurnal cycle. This can be a reason for the relatively large precipitation data-model mismatch for coastal areas where sea breeze can initiate convection.

A note of caution, our downscaling mostly relies on the internal physics of the original ECBilt model. Given the relative simplicity of the scheme, the small scale processes are not explicitly taken into account. As such, the methodology presented here might not be always suitable for high resolution modelling where the small scale processes can become dominant. Also, in our approach, winds are not used for the precipitation distribution within a coarse grid. A foreseen future model development

would be to implement a scheme to increase the precipitation for windward points relative to the leeward ones.

We have shown that the downscaling has only a limited impact on the temperature field at T21 resolution. This is partly due to the fact that the large-scale atmospheric circulation remains mostly unchanged whilst using the downscaling (not shown).

However, at T21 resolution, there are some local changes in precipitation, though these are localised predominately over mountainous areas. Thus, some components of the model, such as continental runoff and ultimately ocean, or vegetation, are impacted by the inclusion of the downscaling. In one simulation of 1,000 years we integrated for one particular parameter combination we obtained a modified state for the ocean and the vegetation. Though the total amount of precipitation in the northern hemisphere is not modified substantially the spatial distribution of the precipitation in the different runoff basins led to a reduction of the Atlantic meridional overturning circulation strength and to a shallower branch of the upper branch of the thermohaline circulation in that particular simulation. To avoid this global climate drift from the CTRL experiment, we present only 100 years of model integration ensuring a limited role of the downscaling feedbacks on the global climate. However, for longer integration, the model might need some adjusment in order to correctly reproduce the present-day state of the climate system.

In an earlier version of the model that included a coupled ice sheet, Roche et al. (2014) demonstrated the poor performance of the surface mass balance when it is computed from bilinearly interpolated precipitation in simulating the present-day topography of the Greenland ice sheet. From the downscaled atmospheric fields shown here, it is now possible to compute the surface mass balance required by the ice sheet model embedded in *i*LOVECLIM. This downscaled surface mass balance will explicitly take into account the sub-grid temperature and precipitation according to the local orography. With this, we aim at better reproducing the non-linear nature of the surface mass balance and in particular the position of the ablation zone at the margin. Foreseen applications include ice sheet - climate interactively coupled thanks to the downscaled atmospheric fields although the artificial discontinuities due to the imprint of the coarse native grid cell in the downscaled field are still an important drawback of the method presented. Ice sheet mass balance is not the only possible application as our methodology is not grid-specific and can be used to compute high resolution temperature and precipitation required for any submodel. Thus, foreseen applications include the computation of high resolution terrestrial water cycle, in particular for permafrost and vegetation dynamics.

## 5   Code availability

The *i*LOVECLIM source code is based on the LOVECLIM model version 1.2 whose code is accessible at http://www.elic.ucl. ac.be/modx/elic/index.php?id=289. The developments on the *i*LOVECLIM source code are hosted at https://forge.ipsl.jussieu. fr/ludus, but are not publicly available due to copyright restrictions. Access can be granted on demand by request to D. M. Roche (didier.roche@lsce.ipsl.fr) to those who conduct research in collaboration with the *i*LOVECLIM users group. For this work we used the model at revision 706.

*Author contributions.* A. Quiquet and D.M. Roche designed the project. D. Paillard and C. Dumas contributed to the discussions on practical implementation. A. Quiquet and D.M. Roche implemented the new functionality in the climate model. A. Quiquet performed the simulations. All authors participated in the analysis of model outputs and the manuscript writing.

*Acknowledgements.* This is a contribution to ERC project ACCLIMATE; the research leading to these results has received funding from the European Research Council under the European Union's Seventh Framework Programme (FP7/2007-2013)/ERC grant agreement 339108. The authors gratefully thank M. Vrac and K. Izumi for fruitful discussions. We acknowledge the Institut Pierre Simon Laplace for hosting the iLOVECLIM model code under the LUDUS framework project (https://forge.ipsl.jussieu.fr/ludus).

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

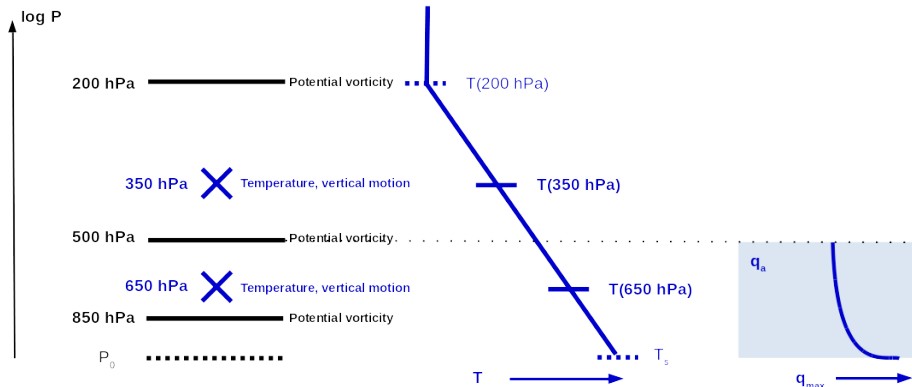

**Figure 1.** Schematic representation of the atmosphere in ECBilt. The three levels for the vorticity equation are 200, 500 and 850 hPa. The temperature is effectively computed for 350 and 650 hPa, and then linearly interpolated on a log scale to any other pressure level. The saturation profile in the moist layer (below 500 hPa) is computed from tabulated values.

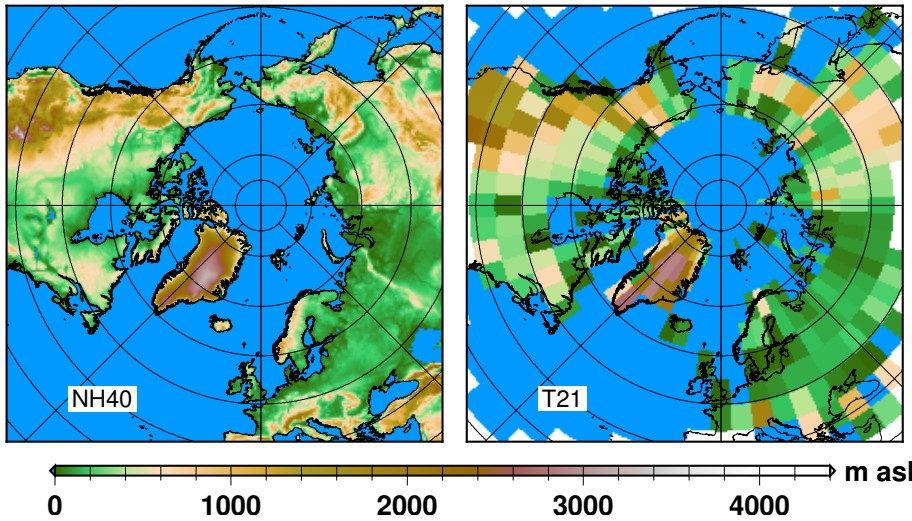

**Figure 2.** Norhern Hemisphere topography from ETOPO1 projected with a Lambert equal area on a Cartesian 40 km by 40 km grid (left) and in the native ECBilt grid (right).

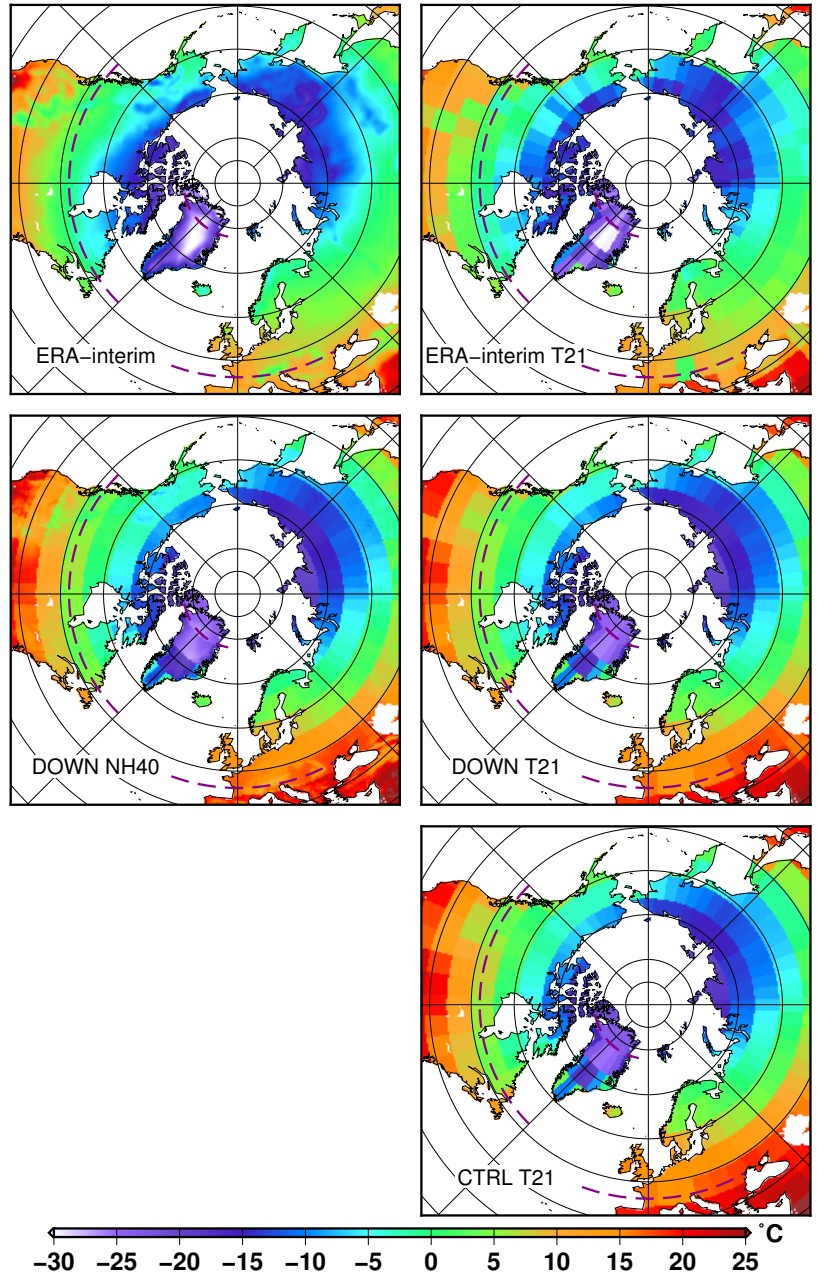

**Figure 3.** Norhern Hemisphere annual mean surface temperature (°C) in: ERA-interim (top), the *i*LOVECLIM that includes a downscaling (middle, with $z_q = 2000m$, $\alpha_q^{min} = 0.8$ and $f_s = 0.6$) and the standard version of *i*LOVECLIM (bottom, CTRL). The left panel corresponds to data on the high resolution grid, whilst on the right data are aggregated to T21 resolution. The dashed purple lines stand for the selected transects used for discussion.

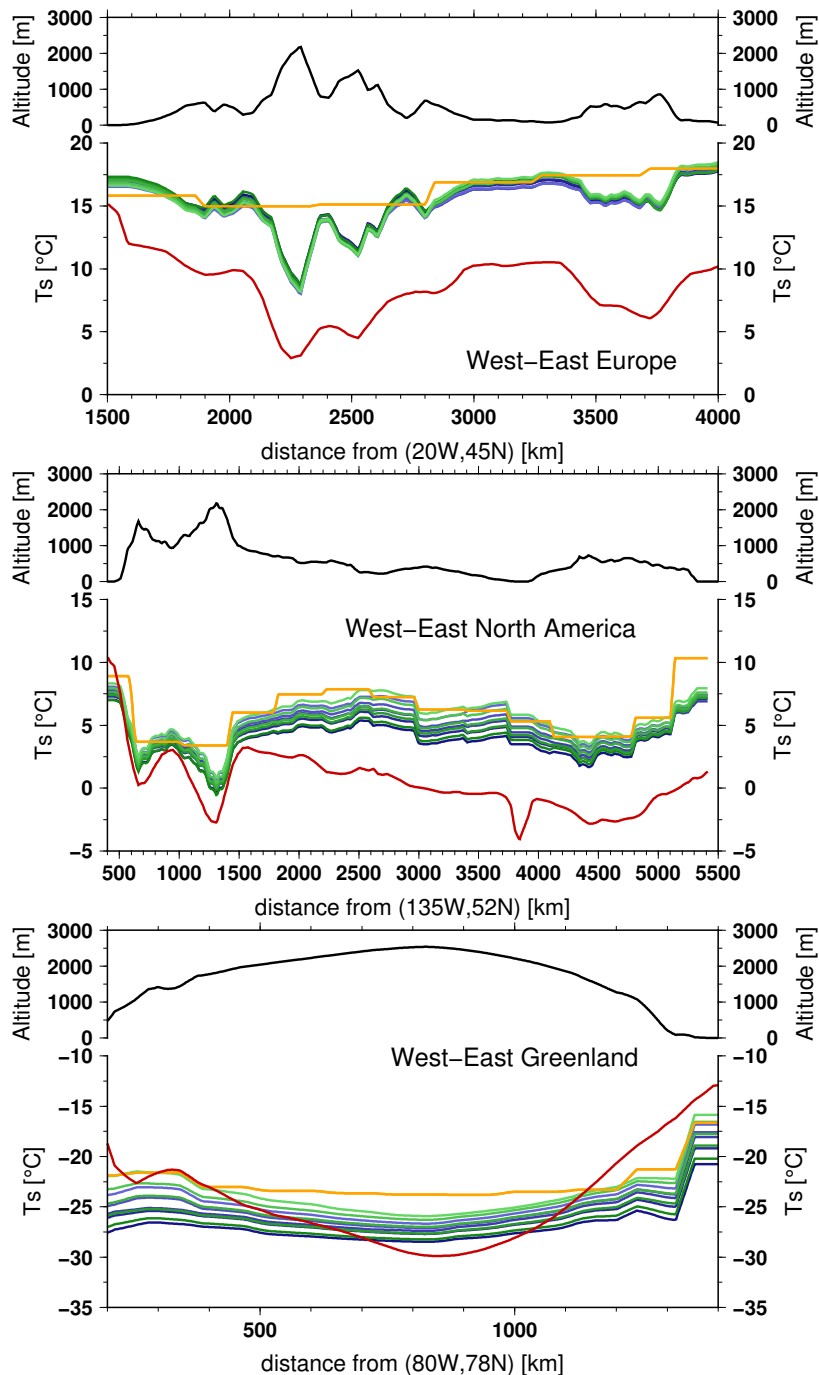

**Figure 4.** Transects for selected regions: Europe (top panel), America (middle panel) and Greenland (bottom panel).The upper part of each panel shows the elevation along the transects.The lower part of each panel depicts the annual mean surface temperature along the transects for: ERA-interim (red), the standard *i*LOVECLIM (CTRL, orange), the *i*LOVECLIM including a downscaling with $f_s = 1.0$ (blue), the *i*LOVECLIM including a downscaling with $f_s = 0.6$ (green). The different shades of blue and green correspond to $\alpha_q^{min}$ ranging from 0.7 (dark) to 0.9 (light). The downscaling experiments presented in this figure use $z_q = 2000m$ and a change to $z_q = 3500m$ has only a very limited effect.

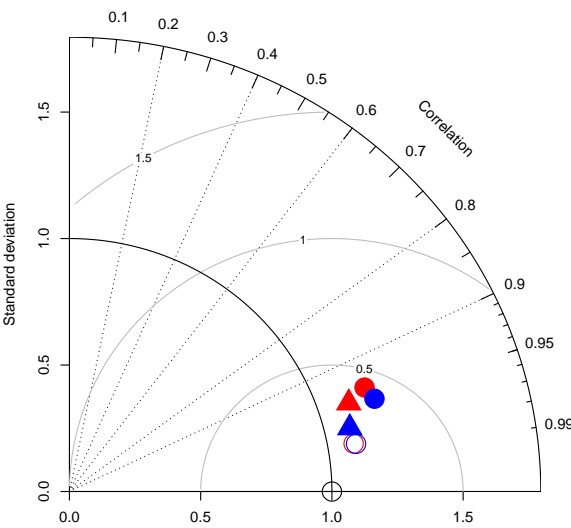

**Figure 5.** Normalised Taylor diagrams on the ERA-interim annual mean surface temperature for the standard CTRL experiment (red) and a selected downscaling experiment (with $z_q = 2000m$, $\alpha_q^{min} = 0.8$ and $f_s = 0.6$) (blue). The circles depict the score when all grid points are considered, whilst the triangles stand for points with an elevation greater than 800 m. The filled symbols correspond to the Taylor Diagram computed on the high resolution grid whilst the symbols outlined-only are for the T21 grid. In this figure, the metrics (standard deviation, correlation and root mean square error) are computed from the annual mean climatic variables. The standard deviation in the observations is used to normalise the standard deviations and the root mean square error.

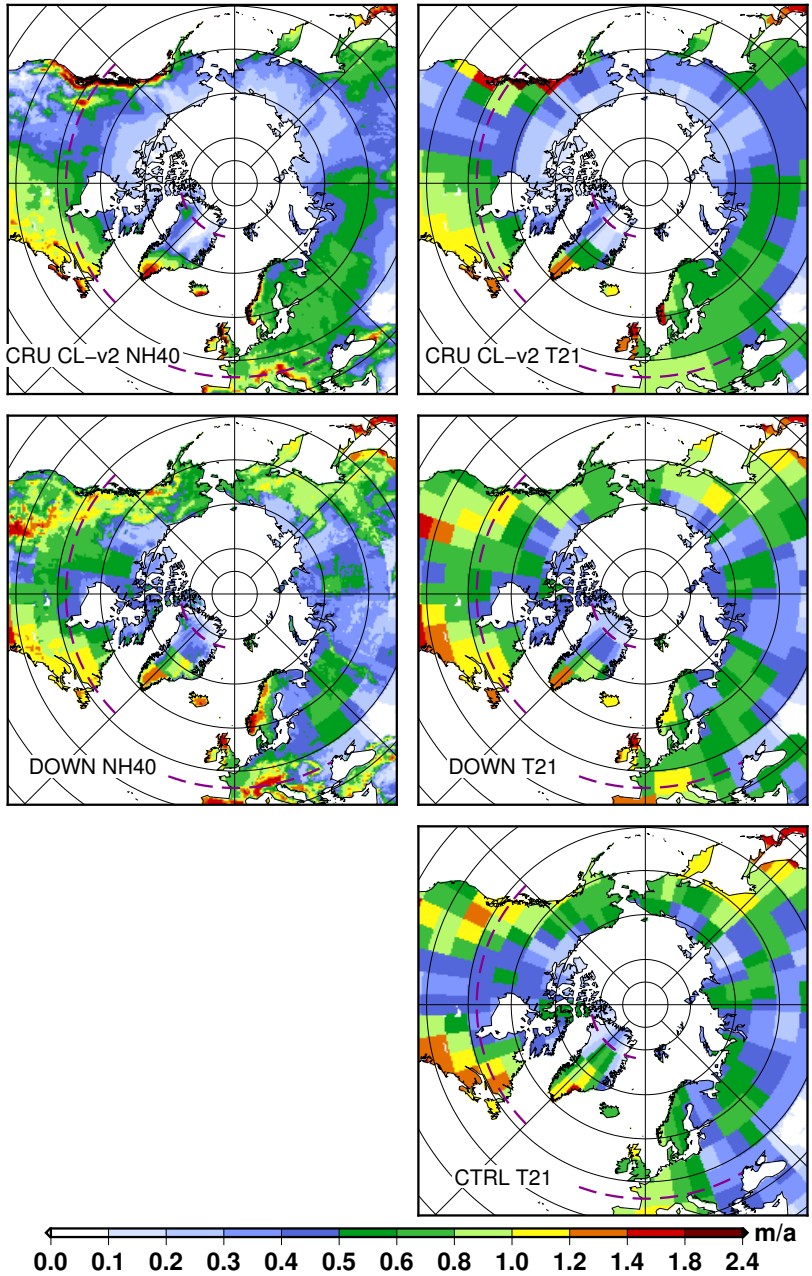

**Figure 6.** Norhern Hemisphere annual mean precipitation rate (m/yr) in: CRU CL-v2 (top), the *i*LOVECLIM that includes a downscaling (middle, with $z_q = 2000m$, $\alpha_q^{min} = 0.8$ and $f_s = 0.6$) and the standard version of *i*LOVECLIM (bottom, CTRL). The left panel corresponds to data on the high resolution grid, whilst on the right data are aggregated to T21 resolution. The dashed purple lines stand for the selected transects used for discussion.

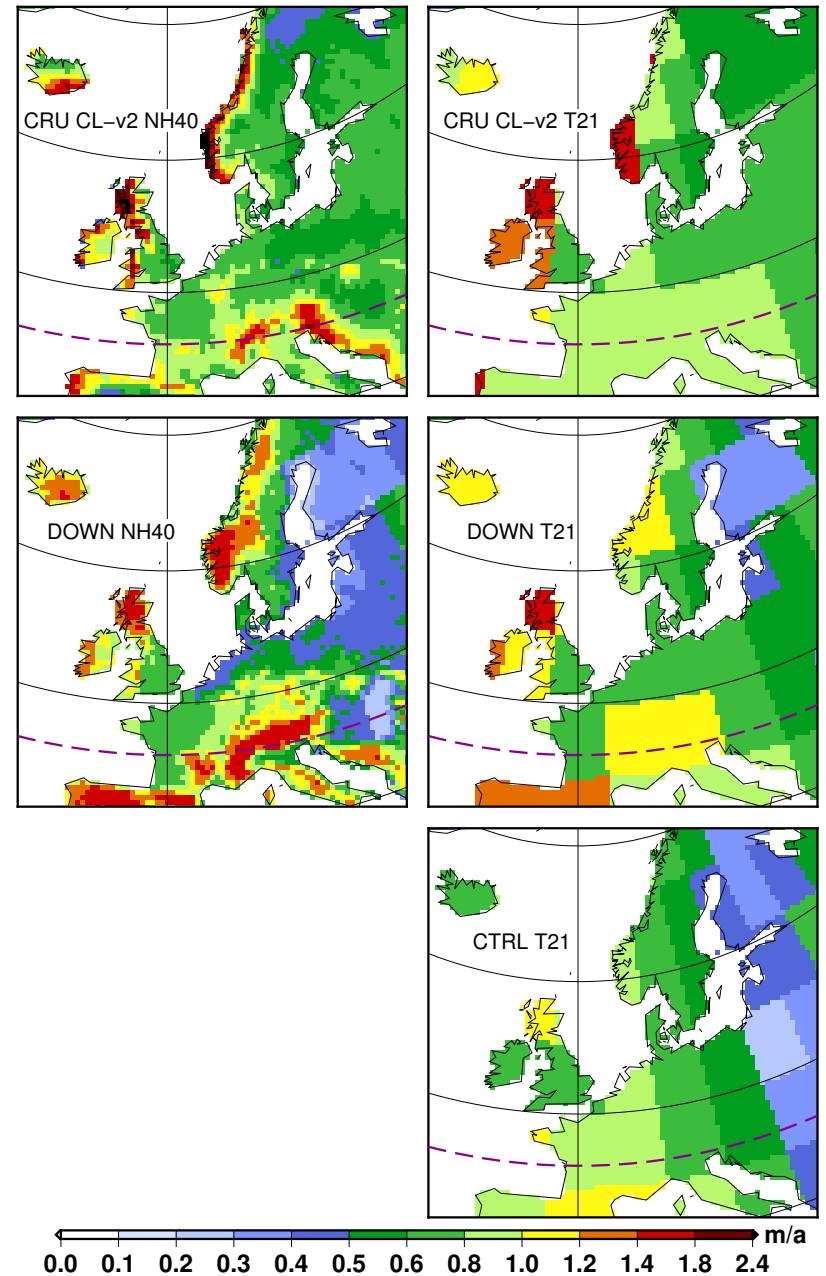

**Figure 7.** Same as Fig. 6 but zoomed over Europe.

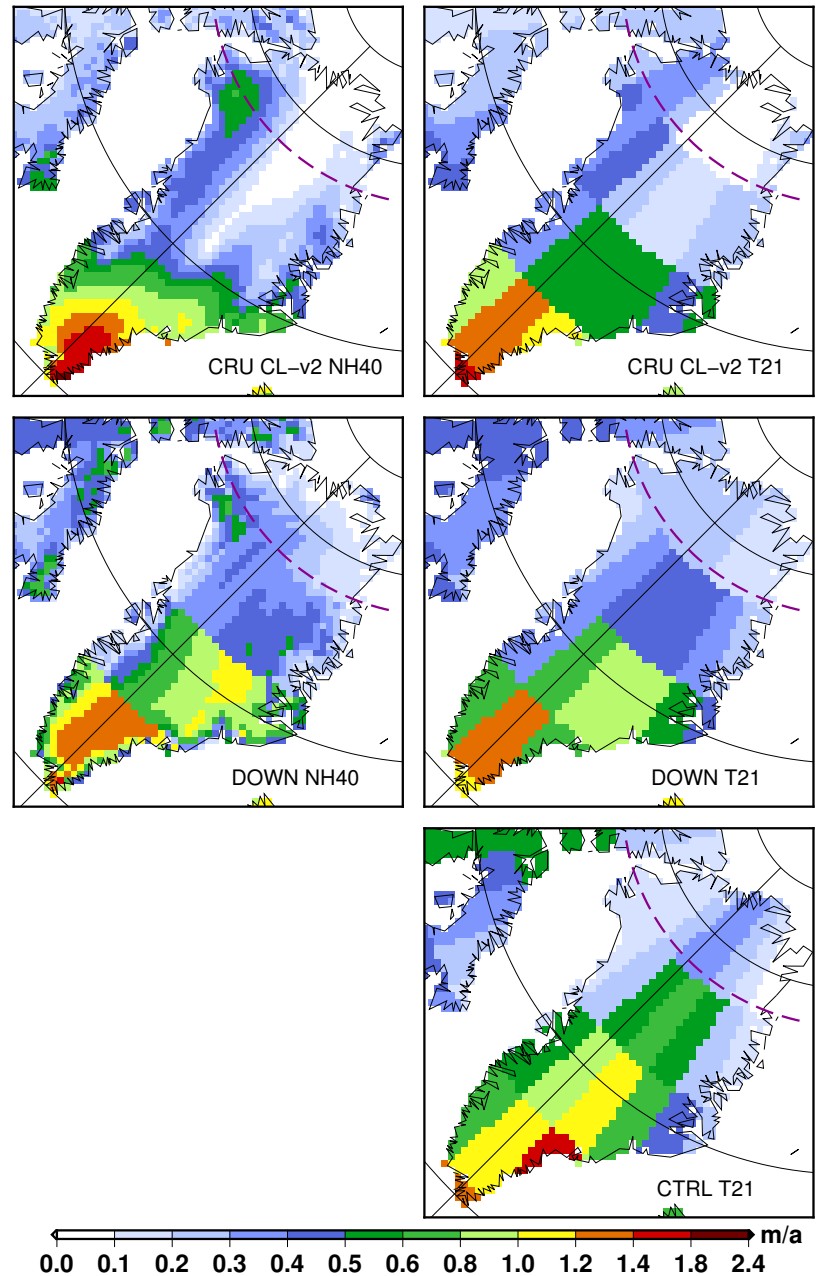

**Figure 8.** Same as Fig. 6 but zoomed over Greenland.

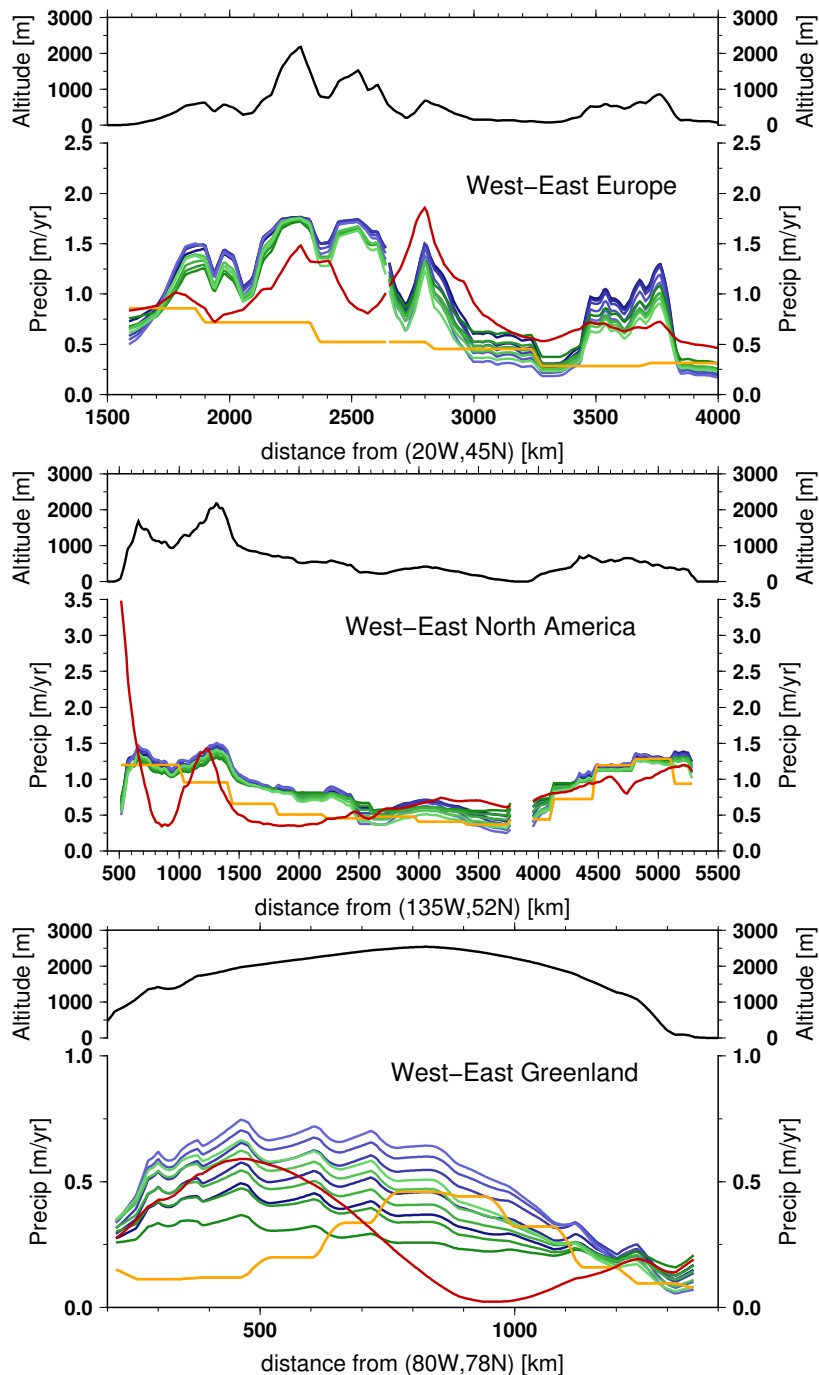

**Figure 9.** Transects for selected regions: Europe (top panel), America (middle panel) and Greenland (bottom panel).The upper part of each panel shows the elevation along the transects.The lower part of each panel depicts the annual mean precipitation along the transects for: CRU CL-V2 (red), the standard *i*LOVECLIM (CTRL, orange), the *i*LOVECLIM including a downscaling with $f_s = 1.0$ (blue), the *i*LOVECLIM including a downscaling with $f_s = 0.6$ (green). The different shades of blue and green correspond to $\alpha_q^{min}$ ranging from 0.7 (dark) to 0.9 (light). The downscaling experiments presented in this figure use $z_q = 2000m$ and a change to $z_q = 3500m$ has only a very limited effect.

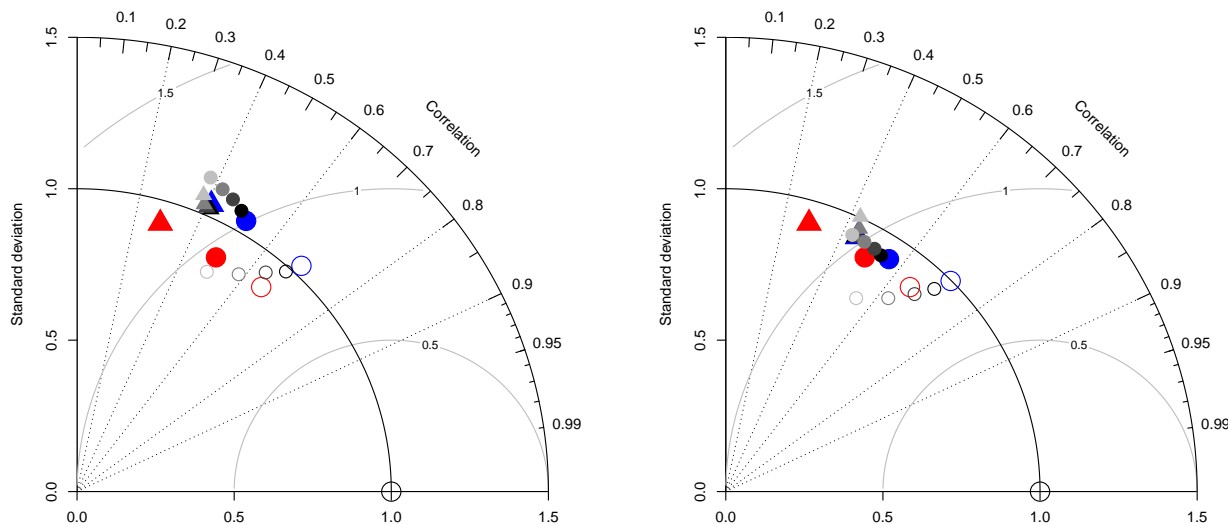

**Figure 10.** Normalised Taylor diagrams on the CRU CL-V2 annual mean precipitation rate for the standard CTRL experiment (red) and a series of DOWN experiments (grey and blue). The circles depict the score when all grid points are considered, whilst the triangles stand for points with an elevation greater than 800 m. The filled symbols correspond to the Taylor Diagram computed on the high resolution grid whilst the symbols outlined-only are for the T21 grid. All the DOWN experiments presented here use $z_q = 2000m$. The different shades of greys are for different $\alpha_q^{min}$ ranging from 0.75 (dark) to 0.9 (light), for $f_s = 1.0$ (left) and $f_s = 0.6$ (right). DOWN with $z_q = 2000m$, $\alpha_q^{min} = 0.7$ and $f_s = 1.0$ (left) and DOWN with $z_q = 2000m$, $\alpha_q^{min} = 0.7$ and $f_s = 0.6$ (right) are in blue. In this figure, the metrics (standard deviation, correlation and root mean square error) are computed from the annual mean climatic variables. The standard deviation in the observations is used to normalise the standard deviations and the root mean square error.

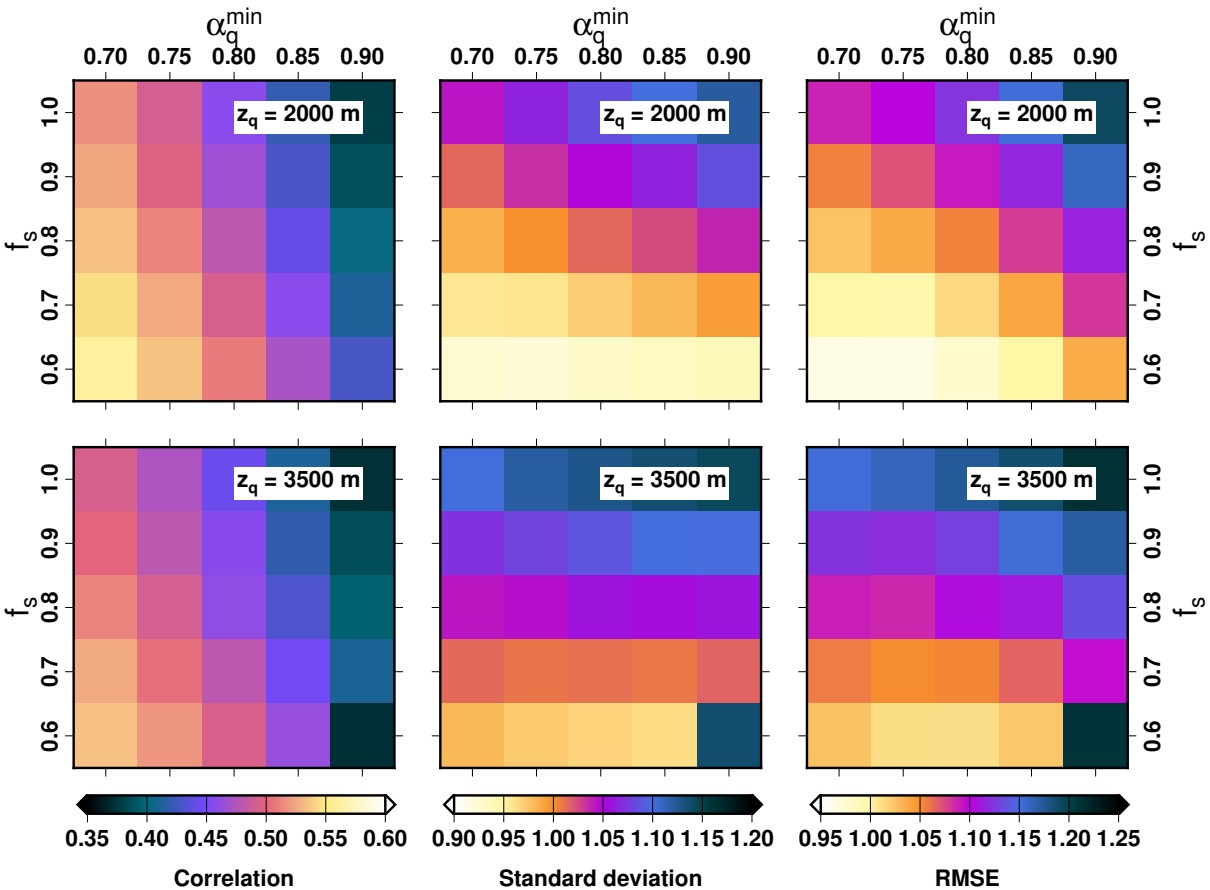

**Figure 11.** Correlation, normalised standard deviation and normalised root mean square error computed from annual mean precipitation as a function of the parameter values for the downscaling experiments. The normalisation is done by dividing the modelled metric (either standard deviation or root mean square error) by the standard deviation in the observations.