# Peer review of "Online dynamical downscaling of temperature and precipitation within the *i*LOVECLIM model (version 1.1)"

_Geoscientific Model Development, 2017_

## Referee Comment (RC1) · Anonymous Referee #1 · 21 Jul 2017

The article describes numerical methods that allow for temperature and precipitation downscaling within the iLOVECLIM model version in an online mode. There is a clear need for an improved spatial representation of these climate variables inside coarse-resolution EMIC models (Ice-sheet modeling is one aspect, but vegetation-climate interactions or forward proxy modeling will clearly benefit from such online downscaling scheme, too). The numerical methods are well-reasoned and certainly make sense within the iLOVECLIM model physical parameterizations. The authors describe their numerical scheme in detail so that it is transparent and can be reproduced or modified by others. The validation or model evaluation is sufficient, but I have a few suggestions to the authors to increase the value of the comparison with the observations (and to

the standard model version). The discussion of the results, the improvements (and lack of) of the precipitation and temperature fields fell a little short, in my opinion. Another interesting aspect would be to discuss how the redistribution of precipitation inside the coarse resolution grid cell can affect the river routing runoff and if that affects in any way the ocean circulation. Therefore related to that question is, to what extent could this method be applied to tropical regions and Antarctica? This should at least be discussed since other users of the model may want a globally applicable downscaling scheme. Ideally this discussion should include a few sentences on the cost of adding additional regions to the downscaling process.

Before I will go into the specific comments and remarks I wanted to point out that the phrase 'dynamical downscaling' is very much restricted in use currently and applies to the application of regional climate models nested within a GCM and /or forced with boundary conditions. Therefore, I would argue against using the term in the title.

Introduction:

p. 1, l. 14-15: This could be extended to include many other applications of EMICs in process studies of the Earth System. Please add a few more examples (in connection with LOVECLIM, e.g. the research labs of Dr. Axel Timmermann, Dr. Hans Renssen, Dr. Andre Berger, and last but not least, Dr. Hugues Goosse have done extensive work with LOVECLIM (and your own research team, too). Likewise Dr. Ganopolski's work deserves to be mentioned, too, in connection with glacial cycles modeling. One could go one with the list, of course and include work of other research teams that apply other EMIC model like the climate modeling group (Dr. Andrew Weaver, Dr. Michael Eby) at University Victoria http://climate.uvic.ca/model/). I leave it to the authors to expand this paragraph in the introduction.

p.1 l. 21: "This has important . . ."

p.2. l.1-2: "high resolution is a particularly dire . . . require high spatial gridding" Aside from being a tautology this sentence needs to be revised carefully. (And note: avoid

use of 'dire' in this context)

p.2 l.5: Your downscaling of temperature and precipitation are first and foremost important for the surface mass balance (SMB) of ice sheets. The grounding-line problem constitutes another 'grid-resolution' problem independent of the SMB. Please explain more carefully how the processes you discuss are physically connected and how your downscaling can help to address specific problems.

p.2. l.10 not sure if the journal has specific grammar rules but I would prefer "another" vs "an other" (here and in other sections of the text)

p.2 l.20 Consider to 'relabel' your downscaling, instead of using the dynamical downscaling, which is for many a term indicating the explicit use of a regional climate model.

2 Methodology

2.1 The iLOVECLIM model

The description should include some description of the 3-dim ocean model, which set's iLOVECLIM apart from other EMICs that use a 2-dim oceans, or slab-ocean-type models. Also, in connection with my comments on discussing the effects of precipitation downscaling on river runoff and routing into the ocean, it would be good to give the reader some brief insight how the ocean is represented in iLOVECLIM.

p.4. l.8-11: Has there been made any attempt to validate this correction factor using ERA interim data, for example? Or could one use the reanalysis data to constrain the correction factor f_s?

p.4 l. 13 "[. . .] we derive several surface energy balance terms [. . .]"

p.4 l. 27 (last sentence) and p.6. l.3-4 and eqn. 5:

I had difficulties to follow the calculations of the moisture profile and the use of the relative humidity profile in the dynamic precipitation calculations. Is the relative humidity iteratively calculated starting with a constant profile in relative humidity? Are you then

updating it to an actual profile that corresponds to the moisture profile after dynamic precipitation was calculated? On page 4 you say relative humidity is constant below 500hPa. On page 6 you diagnose the relative humidity on the virtual levels.

p. 5 l. 6 : write 'area' instead of 'surface': "Where [. . .] is the area of the sub-grid cell."

p.5. l. 16: "[. . .] this approach as computationally too expensive at this time."

p.5.l.18: "initialized with"

3 Application and validation:

3.1.1 Experimental design

p.7 last paragraph: 100 year simulations seem to be rather short for a coupled model. Can you explain what restart state was chosen, and was it really only a 100-yr integration, or did you have a longer spin-up simulation and only analyzed the last 100 model years?

p.8. l.4-5: Interesting point for the application: So right now you have perhaps downscaled less than 40% of the globe, and you have shown that it is most effective in proximity and over land with orographic features. Would it be possible to add more regions (e.g. Antarctica) in parallel and effectively keep the computational costs at similar loads?

p.9. l.17 and l.23: Reading the text up to line 17-18 one wonders what is the reason? Lines 23-24 seem to address the same issue. Consider rewriting this section and discuss the potential reasons.

p.9. l.25: "[. . .] precipitation decrease. Although the Northern [. . .]"

p.9. l.28-29: Please add an explanation. Is it because of your mass-conservation scheme or can in principle the coarse grid cells end up with significantly higher or lower precipitation after the downscaling? (Or did I overlook the text section where you discuss the how numerical downscaling scheme imposes certain constraints on the

area-averaged rainfall).

p.10 line 13 "[...] performance on one specific metric but not the others": "others" or the "other one" In the Taylor diagram there are only two metrics combined.

p.10 l.16 "[...] range tested [not shown]. The real benefit of [...]"

p.10 l. 20-25: This deserves more discussion. How is the long-term simulation affected by the introduced downscaling scheme? In this regards I can think of the ocean-atmosphere interaction, in particular the river routing and runoff into the ocean. Some studies have shown that numerical models can be quite sensitive to a re-routing of freshwater into the ocean. Other implications worth to discuss: how does it affect vegetation cover in the VECODE, and could it potentially lead to feedbacks. Finally, since you started the introduction with references to ice-sheet modeling, it would actually be good to show some example perhaps from Greenland ice sheet model? There you have a significant improvement in the precipitation profile and an effect on the SMB should have an impact on the representation of Greenland's ice sheet.

Also notice that, in the summary on the same page you say "The scheme is conservative and, as such, is suitable for long-term integrations." (l. 31). So, in between these two statements (l.20-25 and l.31) there should be an extended discussion that leads to your concluding statement on l.31.

Figures:

Fig. 2, 5 I would have preferred if the figures showed the following difference maps:

X: stands for the climate variable

M: for model (M_CTLR, M_DOWN)

O: for observational data (reanalysis)

LR, HR: for low and high resolution respectively

Then arrange the figure in the follow 3x2 grid:

left column HR, right column LR

top row: observations O

middle row: M_DOWN

bottom row: M_CTRL

In addition then the corresponding difference maps in a 2x2 grid

left column HR, right column LR

top row: difference M_DOWN - O

bottom row: difference M_CTRL - O

Further suggestions:

Could you mention if/how the large-scale modes of variability in the Northern Hemisphere or the interannual variability are affected by the downscaling? There was only briefly mentioned that the effect on the circulation was small.

Is it worth to report on land model components, such as snow cover, vegetation cover or are there no significant changes?
* * *

---

## Referee Comment (RC2) · Dr. Fyke (Referee) · 25 Jul 2017

This manuscript describes a method to downscale temperature and precipitation from a coarse EMIC grid to a higher resolution grid, such as (for example) that typical for ice sheet models, but also other aspects (e.g. mountainous environments). This is a very pertinent, needed, and surprisingly difficult topic, and it's great the authors are tackling this in iLOVECLIM.

I think some work is still required before this can be published, so suggest a number of revisions. Some general themes that require improvement in my opinion, and that are reflected in my more detailed comments, are:

-Given the 'EMIC' nature of the model and simplifications within the downscaling scheme, I think greater emphasis on caveats associated with EMIC-embedded downscaling should be more clearly described, so that potential users are clearly aware of what aspects of their scientific simulations are a result of model simplifications, versus real processes.

-A clearer high-level, but technical overview of the scheme would be really useful in the Introduction, so the readers enter the details with a pre-existing, rough, mental construct of what to expect. E.g. 'Briefly, the downscaling procedure reimplements the original ECBILT equations at 11 vertical levels, followed by... and...' Also, repeat this overview at the end.

-Several important equations are presented without justification (e.g. 'we used this formulation because...'). Even if Haarsma et al. (1997) is cited, it would still be useful to provide a physical reasoning for the equation form.

-Given the advertised capability of the downscaling, to enable science to be done at sub grid scales (e.g. GrIS ablation zones...) it seems necessary to show some 'zoomed-in' plots of particular regions, for example, GrIS. As it stands, the reader has to squint to try and see the downscaled fields, as they are represented across the whole domain.

-Some evaluation concerns: —>there appears there is a non-monotonic saw-tooth or wave pattern in precipitation across Greenland, generated by the precipitation downscaling. This would be very problematic for actual SMB fields used for science. These features, and other potentially similar 'native grid artifact' features over non-ice-sheet regions (which I am less familiar with) need to be explained in more detail. ->the authors note that downscaling appears to have a significant effect on the continental-scale distribution of precipitation at the native grid scale but don't clearly discuss why. Deeper discussion on this topic would be excellent to see.

Comments:

[Figure]

P1L20: "approximative representation of land surface climatic variables that are affected by variability at high spatial resolution." Confusing, suggest more clearly indicating that by variability the authors mean spatial variability that is lost at low resolution.

P1L24: "EMICs are unable by design to reproduce correctly meso-scale atmospheric processes induced by sub-grid topography": suggest just saying "relatively fine-scale topographic features such as mountain ranges"

P2L2: "physical description require a high spatial gridding.": suggest: "large-scale physical behavior depends highly on processes occurring at small spatial scales"

P2L8: "superimposed to" -> "added to".

P2L9: "Such a strategy implicitly assumes that the model biases remain unchanged through time, independently from the imposed external forcings." Also, that biases remain unchanged as ice sheet geometry changes significantly.

P2L14: "Fyke et al. (2011) go a step further as not only temperature but also precipitation". Technically, we didn't actually go the step farther: only temperature is downscaled in Fyke et al (2011). However, this is used in the energy balance and precipitation calculations, rain/snow decision, and relative humidity, which impacts the decision on whether to precipitate or not. But, precipitation is not downscaled directly, per se.

P2L20: Importantly, I don't think the method is downscaling heat. It is downscaling temperature - which is not the same as heat.

P2L25: "closed water budget" -> "closed global water budget"

P2L29: The scheme could also be important for dynamic vegetation models (i.e. by resolving sub gridded elevation-dependent vegetation distribution envelopes)

P3L23: Review the justification for basing the linear temperature profile on the log of pressure (even if already stated in Haarsma et al (1997) and/or basic knowledge)

P4L1: why is the interpolated T500 used to obtain the near-surface temperature, instead of T650 (which is presumably closer to the surface)?

P4L1: Explain why different treatments are applied to near-surface temperature, versus T(p). Does this result in a discontinuity in temperatures, when comparing near-surface temperatures to surface temperatures? Perhaps I'm just confused here.

P4L9: "Due to orography, the atmospheric isotherms are shifted upwards": is there a citation explaining this physical phenomenon? Given it's role in motivating implementation of the f_s factor, it seems important to explain.

Equations 3 and 4: nearly duplicative. Could the authors just present Equation 4, then say "f_s=1, reduces to the original equation of Haarsma et al. (1997)"?

Equations 3 and 4: why is T* over-barred in Eq 3, but not in Eq 4?

P4L13: For completeness, perhaps explicitly state these energy balance terms, indicating which ones use as input the downscaled near-surface air temperature.

P4L19: "In ECBilt,..." -> "In the idealized ECBilt representation of the atmosphere,..."

P4L20: What are the actual z_h heights? I don't actually see them specified anywhere in the manuscript...

Equation 5: could g be taken out of the integral (also, in the original Haarsma equation)?

General: suggest including web link for Haarsma et al. (1997) in reference, as it is not available by, e.g. DOI. As it is it takes some brief Google searching to find a PDF copy.

General: how does scheme work for elevations greater than 5500m (500 hPa)? Since mountainous regions are of specific interest, and many of these regions have elevations greater than 5500m, this would seem important to note.

Equation 6: it's not immediately clear why the authors need to calculate surface pressure as a function of surface temperature? Can't the authors apply a more direct pressure/elevation relationship? If I'm wrong (quite possibly) - perhaps a clearer description of why this equation form is used, could be useful.

General: the switching between use of values at the 650 hPA and 500 hPa levels is somewhat confusing. Can motivation/clarification be given on why this switching occurs?

Equation 7: is $s_a(k)$ 'surface' or 'surface area'?

Equation 7: I'm not sure the initial $1/k_{max}$ term is correct here.

Equation 7: the authors could just say 'the surface elevation of the native grid comprises the area-weighted average of all k sub-grid points.', could they not?

Equation 8: I'm not sure it's necessary to write out the equation for linear interpolation here. I'd be OK with just saying 'linearly interpolate from the bounding vertical levels' or something similar.

General: some devil's advocate points that might be worth addressing: why not just compute $T^*$, $T_s$, and $q_{max}$ at each sub grid point, using the equations described earlier? What is the advantage of first calculating these at specified levels, then vertically interpolating? Related: others (e.g. in Fyke et al. (2011) didn't use 11 globally constant levels, but rather calculated the range of levels at each point, based on the high resolution topography within each native grid cell. This significantly reduced computation #s (for example, over large flat regions) and also allowed for finer vertical resolution in the sub gridded levels. Was this approach considered here?

P5L17: how computationally expensive would it be, really, to bin sub grid points by aspect (in relation to wind direction), especially in the context of the full coupled model cost?

P5L20: so, for leeward slopes, this means that the lowest leeward slopes receive more than high leeward slopes (and leeward slopes get the same precip as same-elevation windward slopes). These should be explicitly discussed, as a representative 'caveat'

of the scheme.

P5L20: also, in my understanding, it is 'mid-level' windward slopes that often get the most precipitation. Whereas the described scheme bias precipitation to low levels. Another caveat to emphasize.

Section 2.3.2: 'Dynamic precipitation' is an unknown term to me (a non-meteorologist). If it isn't a common meteorological term, perhaps re-name, or explicitly define.

P5L26: if the local topography exists above 500 hPA, does it receive any precipitation at all in the model?

P5L27: "expended"->"expanded"

Equation 12: it's not clear what the physical justification is for the form of this equation. Please describe in greater detail, with citations.

General: which of the 3 contributors to precipitation (2x dynamic, and convective) is generally the dominant term? This would be useful for readers to know.

P7L2: It's not clear to me how the convective precipitation scheme works. For example, given the repeated use of Eq. 13, where does the assessment of stability come in? I think a clearer description is needed here.

P8L2: by 40%, for the whole coupled model? Or just the atmospheric component?

Section 3.1.1/3.2: note the caveat that the authors are comparing a preindustrial simulation to recent historical climatologies (or describe why this isn't a caveat, e.g., why recent historical climatology is close enough to preindustrial climatology for the fields in question, to warrant direct comparison)

P7L16: given 'continentality' is usually associated with sub-annual ranges in temperatures, what does it mean to interpret 'continentality' over Siberia, when using annual-mean fields? Furthermore, it is unclear how this relates to other regions (as it is written, it seems to indicate that increased Siberian continentality causes biases elsewhere)?

P7L18: "does not imply important changes in surface temperature": relative to the default CTRL case? Perhaps reword for clarity.

P7L30: Given the 'ijk' indexing is hardly used in the manuscript (as figures mostly show the results only from one ensemble), I'm wondering how useful it is to describe this indexing scheme? It would become more useful, if plots of results as a function of parameter space, were shown. . .

Figure 2/4/5/6: why was DOWN020 chosen as the representative plot? How much do the different ensemble members look, and why/why not?

Figure 2, 'DOWN NH40' panel: it is surprising to see remnants of the native grid in many places, though I suspect I know why. I think a description of why this remaining imprinting occurs should be clearly explained to readers.

Figure 3: The green/blue shading is quite confusing to parse, visually. Could shaded 'clouds' be more visually accessible?

P8L26: ". . .to correct the model bias. . ." -> ". . .to correct broader region model biases that are unrelated to topographic forcing. . ."

P8L27: ". . .horizontal gradients. . ."->". . .horizontal temperature gradients. . ."

General: given the advertised ability of the scheme to downscale high-resolution map-view T/P fields, and the intention of the authors that the downscaling will improve 'regional' studies, it would be useful to see regional subset plots of Figure 2, ERA-interim and DOWN NH40. For example, given my background, I'd like a closer look at GrIS precipitation!

Figure 4: A better description of the Taylor diagram scheme would be good. For example, is 'standard deviation' the standard deviation in mean annual values, for the long-term climatology (or is it describing the standard deviation in temporal variability?). Similarly for the correlation axis.

P9L8: describe why the lack of impact on native-grid model performance is a good/bad thing.

General: a more robust description of the analysis of the full 50-member analysis is warranted. For example, which varied parameter makes the most difference? Is it possible to identify parameter combinations that are optimal, for particular locations?

Figure 5: contrary to the text, it looks to me like DOWN NH40 *does* better resolve the Norwegian/W. North America high-precipitation bands. . .

General: A stronger justification is needed that downscaling does indeed produce 'scientifically useful' high-resolution precipitation fields. As it is, the reader is somewhat left to their own devices to piece together the various impacts of precipitation downscaling into a coherent story on how well the precipitation downscaling scheme (perhaps the most important but tenuous aspect of the whole procedure) performs, and whether it would help/hinder their scientific simulations.

Figure 6: The GrIS cross section highlights some concerning 'saw-tooth/wave' behavior, whereby the downscaled precipitation field changes across the boundaries of the native grid. To be honest, we have struggled with a similar (?) thing in CESM, and I ended up putting together this Google-based schematic that tries to explain our particular problem: https://docs.google.com/presentation/d/1gyaIZ5ypZ3XWxf2VTThuBmkpL9Qf18Qg5w_qpbzPm6s/edit#slide=id.p Non-monotonic SMB fields that could certainly overwhelm the positive impacts of downscaling over GrIS, and preclude use of the downscaling scheme for science, where the GrIS-wide SMB field is important. Please comment on why the 'saw-tooth/wave' pattern occurs in the downscaling, and why the authors consider it acceptable for subsequent science using downscaled SMB fields.

Figure 7: as with figure 4: a better description and interpretation of the Taylor diagram would seem important. Also, it's quite hard to pick out salient differences and understand them in a broader context, given the selection of what seems like an arbitrary

subsample the whole ensemble.

Figure 7: it's not clear now the 'spatial correlation is greatly improved', via inspection of the Taylor diagrams. Which dots should the reader compare, to see this impact? Perhaps this is just my eternal personal struggle with Taylor diagrams, though :).

Figure 4: why not use the same set of ensemble members, as in Figure 7 (for consistency, and perhaps just to show that temperature downscaling is not as parametrically sensitive as precipitation downscaling)?

P10L20: Yes, it is notable how downscaling significantly impacts precipitation on the native grid (e.g. CTRL T21 vs. DOWN T21). For example, it appears North America as a whole receives quite a bit more precipitation when downscaling is utilized. Yet this important aspect is not clearly mechanistically explained. A physical reasoning behind this non-negligible impact should be described in the manuscript.

Section 4: I think noting some of the caveats of the scheme would be appropriate to clearly reiterate in this section.

P11L13: Given the advertised importance of downscaling for iLOVECLIM simulations of Greenland, again it would seem appropriate to show 'zoomed-in' downscaled T/P (or better yet, actual SMB?) over Greenland.

P11L13: Is the iLOVECLIM SMB energy-balance-based? It seems all the ingredients are available, at least on the 'l' levels. If PDD-based, conversely, I'm not sure the authors can claim anymore to be globally conservative, since the empirical nature of the energy flux calculations in PDD schemes does not track conservation (unlike EBM models, which are premised on a balance of actual energy fluxes)

General: I think it would be very useful to technically contrast this subgridding scheme with other previously published schemes, to allow readers some greater contrast on similarities/differences.

General: provide a brief overview of the technical stages of the scheme, either at the

start, or the end, of the manuscript. As it stands it's rather easy to get lost in the details.

General: I think some plots of characteristic T/P/precipitation vertical profiles would be very useful for the reader, to see the equations 'put into action' for some representative cases. As it is, I had to spend some time at the whiteboard to get a sense of what the equations actually produced, in terms of actual profiles.

————————————————————

---

## Author Comment (AC1) · 13 Oct 2017

We warmly thank the anonymous reviewer #1 and Dr. Jeremy Fyke (reviewer #2) for their insightful comments. We did our best to follow the suggestions which greatly improved the manuscript to our opinion.

In the following, we reply point by point to each individual comment (referee's comments are italicised). Following our responses, please find our new manuscript in which we highlighted changes from the original version.

**Anonymous Reviewer #1**

The article describes numerical methods that allow for temperature and precipitation downscaling within the iLOVECLIM model version in an online mode. There is a clear need for an improved spatial representation of these climate variables inside coarse resolution EMIC models (Ice-sheet modeling is one aspect, but vegetation-climate interactions or forward proxy modeling will clearly benefit from such online downscaling scheme, too). The numerical methods are well-reasoned and certainly make sense within the iLOVECLIM model physical parameterizations. The authors describe their numerical scheme in detail so that it is transparent and can be reproduced or modified by others. The validation or model evaluation is sufficient, but I have a few suggestions to the authors to increase the value of the comparison with the observations (and to the standard model version). The discussion of the results, the improvements (and lack of) of the precipitation and temperature fields fell a little short, in my opinion. Another interesting aspect would be to discuss how the redistribution of precipitation. Therefore related to that question is, to what extent could this method be applied to tropical regions and Antarctica? This should at least be discussed since other users of the model may want a globally applicable downscaling scheme. Ideally this discussion should include a few sentences on the cost of adding additional regions to the downscaling process.

We have added several figures (Fig.1, Fig.7, Fig. 8 and Fig. 11 in the revised manuscript) and expanded the discussion. We notably included a schematic representation of the different levels in the atmosphere intended at a better understanding of the equations presented (Fig. 1), a zoom of precipitation maps over Europe and Greenland (Fig.7 and Fig. 8) and correlation, standard deviation and root mean square error as a function of sensitivity parameters for precipitation (Fig. 1). We hope that these additions are useful for the description of the results and answer the concerns expressed by both reviewers.

Regarding the general importance of the downscaling for the global climate through the river routing and changes in vegetation, we think that this discussion is overall too broad for the scope of the present manuscript. The aim of our paper is to describe the physical reasoning and its implementation in the model. The discussion of its long term impact on the global climate has more to do with climate dynamics. We nonetheless added the information asked by the reviewer regarding the Atlantic overturning circulation which, as a large scale integrated parameter, does not require lengthy developments on the geographical distribution of each regions (see our response to your later comment on page 5). A larger (in scope) discussion of the runoff and vegetation impacts is foreseen as detailed applications in the future.

Before I will go into the specific comments and remarks I wanted to point out that the phrase 'dynamical downscaling' is very much restricted in use currently and applies to the application of regional climate models nested within a GCM and /or forced with boundary conditions. Therefore, I would argue against using the term in the title.

To our knowledge, the methodologies used for the downscaling of climatic fields of climate models broadly fall in two categories: statistical downscaling or dynamical downscaling. Inside the second category, there is a variety of approaches. As the reviewer mentions, an important community works actively on the use of a regional climate model forced at its boundary by a coarser resolution GCM, either for offline or online applications. However, there are other approaches, such as the use of stretchable grid within the same model in order to zoom over a specific region of interest (e.g. Hourdin et al. 2006). From our point of view, our scheme belongs clearly to the "online dynamical downscaling" even though it does not use any regional climate model. It does not belong to the "zoom" category since we do not recompute the whole dynamics on the sub-grid nor does it changes its own atmospheric grid locally. The sole labelling "downscaling" is not precise

enough to describe our work, as it might also refer to simple interpolation techniques in which the climatic fields are not recomputed in a physically consistent manner. We thus intend to keep the formulation as it is but are very open to an alternative precise formulation in case the reviewer or editor have a better suggestion.

**Introduction:**

p. 1, I. 14-15: This could be extended to include many other applications of EMICs in process studies of the Earth System. Please add a few more examples (in connection with LOVECLIM, e.g. the research labs of Dr. Axel Timmermann, Dr. Hans Renssen, Dr. Andre Berger, and last but not least, Dr. Hugues Goosse have done extensive work with LOVECLIM (and your own research team, too). Likewise Dr. Ganopolski's work deserves to be mentioned, too, in connection with glacial cycles modeling. One could go one with the list, of course and include work of other research teams that apply other EMIC model like the climate modeling group (Dr. Andrew Weaver, Dr. Michael Eby) at University Victoria http://climate.uvic.ca/model/). I leave it to the authors to expand this paragraph in the introduction.

We agree that the first version of the manuscript did not provide sufficient background information on EMICs. As suggested, we added a synthetic review of EMICs abilities and limitations, including a wider, but by no means complete, literature reference:

"EMICs have been initially developed as computationally cheap alternatives to general circulation model especially in the context of studying the role of orbital and carbon dioxide forcing and feedback within the context of glacial-interglacial cycles (e.g. Weaver et al., 1998; Berger et al., 1998; Ganopolski et al., 1998). The addition of interactive ice sheets models allowed for the study of ice sheet dynamics in term of retreat, advance and stability as a key component of the climate system (e.g. Calov et al., 2002; Huybrechts et al., 2002; Charbit et al., 2005). Also, some EMICs include an interactive carbon cycle which allows the investigation of the mechanisms behind the atmospheric carbon dioxide fluctuations during the Quaternary (e.g. Brovkin et al., 2007; Ridgwell and Hargreaves, 2007; Bouttes et al., 2011). With the increasing computing facilities, the EMICs are generally becoming more comprehensive than they used to be. From zonally averaged atmosphere or ocean (e.g. Gallée et al., 1992; Petoukhov et al., 2000), they now often include a three dimensional ocean (e.g. Edwards and Marsh, 2005; Weaver et al., 2001). The atmospheric component has remained a simplified component in EMICs even though they may be sometimes three dimensional but with only a limited number of vertical levels and slightly simplified base equations (e.g. Goosse et al., 2010)."

p.1 l. 21: "This has important : : :"

Done.

p.2. I.1-2: "high resolution is a particularly dire : : : require high spatial gridding" Aside from being a tautology this sentence needs to be revised carefully. (And note: avoid use of 'dire' in this context)

Replaced by "High resolution is necessary for components whose large-scale physical behavior depends highly on processes occurring at small spatial scales".

p.2 I.5: Your downscaling of temperature and precipitation are first and foremost important for the surface mass balance (SMB) of ice sheets. The grounding-line problem constitutes another 'grid-resolution' problem independent of the SMB. Please explain more carefully how the processes you discuss are physically connected and how your downscaling can help to address specific problems.

Independently from the SMB, ice sheet models need a high resolution because of the grounding line instability. We did not want to establish a direct connection between ice sheet mechanics and SMB. We realise that the sentence was confusing and we now simply mention the SMB: "In particular, ice sheet models need a high resolution to account for narrow ablation zones at the margins (Ettema et al. 2009)."

p.2. I.10 not sure if the journal has specific grammar rules but I would prefer "another" vs "an other" (here and in other sections of the text)

Done.

p.2 I.20 Consider to 'relabel' your downscaling, instead of using the dynamical downscaling, which is for many a term indicating the explicit use of a regional climate model.

Please see our previous comment (page 1-2 in this document).

2 Methodology

**2.1 The iLOVECLIM model**

The description should include some description of the 3-dim ocean model, which set's iLOVECLIM apart from other EMICs that use a 2-dim oceans, or slab-ocean-type models. Also, in connection with my comments on discussing the effects of precipitation downscaling on river runoff and routing into the ocean, it would be good to give the reader some brief insight how the ocean is represented in iLOVECLIM.

There is in fact now a few EMICs that have a 3D ocean (e.g. GENIE, Uvic, LOVECLIM). We added the following information on CLIO:

"The LOVECLIM family models contain a free surface ocean general circulation model with an approximately three degrees spatial resolution resolution and 20 vertical layers. It is coupled to a thermo-dynamical sea ice model operating on the same spatial grid."

p.4. I.8-11: Has there been made any attempt to validate this correction factor using ERA interim data, for example? Or could one use the reanalysis data to constrain the correction factor f\_s?

Indeed, this was the original idea we had in mind. However, the computation of an "actual" value computed from re-analysis or regional model is far from being straightforward: it is highly variable in time and space. In addition, given the relatively low horizontal grid resolution we believe that the "actual global" value computed from re-analysis or regional model might not be suitable for our model. This is why we decided to have this parameter as a tunable parameter though being physically founded.

p.4 l. 13 "[: : :] we derive several surface energy balance terms [: : :]"

Done.

**p.4 I. 27 (last sentence) and p.6. I.3-4 and eqn. 5:**

I had difficulties to follow the calculations of the moisture profile and the use of the relative humidity profile in the dynamic precipitation calculations. Is the relative humidity iteratively calculated starting with a constant profile in relative humidity? Are you then updating it to an actual profile that corresponds to the moisture profile after dynamic precipitation was calculated? On page 4 you say relative humidity is constant below 500hPa. On page 6 you diagnose the relative humidity on the virtual levels.

For a specific spatial location, the total precipitable water  $(q_a)$ , the relative humidity (r) and the saturation specific humidity  $(q_{max})$  are vertically integrated values. As such they indeed do not vary on the vertical. However, on the sub-grid the vertical integration will depend on the sub-grid elevation and this is why we have for example a different relative humidity on the vertical levels. We realise that this is somehow confusing when we try to be more specific in the text and we added the following on page 6:

"[...] with r the relative humidity. For a given grid point, the relative humidity shows a constant vertical profile. However, its value depends on the local topography since its computation is derived from the vertically integrated saturated specific humidity (Eq. 5):"

In addition, we also add a figure showing a schematic representation of the atmosphere in *i*LOVECLIM which provides more insight on the model (Fig. 1 in the revised manuscript).

p. 5 l. 6 : write 'area' instead of 'surface': "Where [: : :] is the area of the sub-grid cell."

Done.

p.5. I. 16: "[: : :] this approach as computationally too expensive at this time."

Done.

**p.5.I.18: "initialized with"**

Done. For consistency with the general British English we used "initialised with".

**3 Application and validation:**

**3.1.1 Experimental design**

*p.*7 last paragraph: 100 year simulations seem to be rather short for a coupled model. Can you explain what restart state was chosen, and was it really only a 100-yr integration, or did you have a longer spin-up simulation and only analyzed the last 100 model years?

All the 100 year simulations used a restart from a long (multi-millenial) standard spinup with preindustrial forcing. This is now clarified in the manuscript (p. 8 l. 22-23):

"For model evaluation, we define a control simulation (hereafter CTRL) as a 100 years of *i*LOVECLIM integration under constant pre-industrial external forcing, branched to the standard long-term equilibrated pre-industrial restart"

We intentionally decided to perform 100 year long simulations to avoid potential feedbacks on the other components. We address this point further in this document when we reply to your comment on feedbacks (page 5-6 in this document).

p.8. I.4-5: Interesting point for the application: So right now you have perhaps downscaled less than 40% of the globe, and you have shown that it is most effective in proximity and over land with orographic features. Would it be possible to add more regions (e.g. Antarctica) in parallel and effectively keep the computational costs at similar loads?

Yes. In fact, we let the possibility in the code to mask flat regions. The maximum altitude difference between sub-grid points in a specific coarse grid point is calculated. If this difference is lower than a given threshold (set to 0 for the presented experiments), we do not apply the downscaling to these sub-grid points. In doing so, we can significantly reduce the computational time by masking wide flat regions (e.g. ocean). It would be theoretically possible to distribute different "zooms" over the globe on different computing cores and doing them in parallel, however this is not yet implemented in our model.

p.9. I.17 and I.23: Reading the text up to line 17-18 one wonders what is the reason? Lines 23-24 seem to address the same issue. Consider rewriting this section and discuss the potential reasons.

We rewrite and reorganise this section.

p.9. I.25: "[: : :] precipitation decrease. Although the Northern [: : :]"

**Done.**

p.9. *I.28-29:* Please add an explanation. Is it because of your mass-conservation scheme or can in principle the coarse grid cells end up with significantly higher or lower precipitation after the downscaling? (Or did I overlook the text section where you discuss the how numerical downscaling scheme imposes certain constraints on the area-averaged rainfall).

As correctly noted by the reviewer, the imprint of the moisture availability in the model is relatively conservative. There is nothing in our approach that guarantee it to be always so, but in practice the large scale structures are generally stable and, as a result, the large biases in the model remain. This means that the amount of precipitation in our model is still governed by large scale variables and that the first order effect of the downscaling is to redistribute the precipitation according to the topography (in a physically consistent way). As such, the major model biases are conserved. Interestingly, we have a relatively small change in the total amount of precipitation: the 30N to 90N average value of precipitation is only decreased by 2% when using the downscaling (for the experiment presented in the 2D maps). There is in fact a compensating effect between the increase in precipitation over elevated areas and a decrease over lowland areas. We add this information at the end of this paragraph (Page 10 I31-34):

"This means that the model large scale structures are generally stable and are only slightly impacted by the downscaling. In fact, the first order effect of the downscaling is to redistribute the precipitation according to the topography in a physically consistent way. In fact, there is only a

relatively small change in the total amount of precipitation when using the downscaling as the 30N to 90N averaged precipitation in the experiments presented in Fig. 6 is only decreased by 2% in this case."

p.10 line 13 "[: : :] performance on one specific metric but not the others": "others" or the "other one" In the Taylor diagram there are only two metrics combined.

In the Taylor diagram, there are three metrics combined (spatial correlation, standard deviation and root mean square error).

p.10 l.16 "[: : :] range tested [not shown]. The real benefit of [: : :]"

**Done, thank you for the suggestion.**

p.10 I. 20-25: This deserves more discussion. How is the long-term simulation affected by the introduced downscaling scheme? In this regards I can think of the ocean atmosphere interaction, in particular the river routing and runoff into the ocean. Some studies have shown that numerical models can be quite sensitive to a re-routing of freshwater into the ocean. Other implications worth to discuss: how does it affect vegetation cover in the VECODE, and could it potentially lead to feedbacks. Finally, since you started the introduction with references to ice-sheet modeling, it would actually be good to show some example perhaps from Greenland ice sheet model? There you have a significant improvement in the precipitation profile and an effect on the SMB should have an impact on the representation of Greenland's ice sheet.

Also notice that, in the summary on the same page you say "The scheme is conservative and, as such, is suitable for long-term integrations." (I. 31). So, in between these two statements (I.20-25 and I.31) there should be an extended discussion that leads to your concluding statement on I.31.

You are right that the perturbation of the water cycle due to the introduction of the downscaling will have potentially important feedbacks on the other components, such as the ocean and the vegetation but also on the atmospheric temperature due to the change of the moisture radiative effect. In one simulation of 1,000 years we integrated for one particular parameter combination we obtained a modified state for the ocean and the vegetation. Though the total amount of precipitation in the northern hemisphere is not modified substantially (cf. Page 4-5 of this document) the spatial distribution of the precipitation in the different runoff basins led to a reduction of the AMOC strength and to a shallower branch of the upper branch of the thermohaline circulation in that particular simulation. The climate state obtained after 1,000 years is however still far from equilibrium and thus not suitable for definitive conclusions. One consequence is that one should ideally retuned the model with the downscaling under present-day conditions since the model has been originally tuned with the standard precipitation field at T21 resolution. The aim of the paper is to present the scheme from the atmospheric point of view, but its actual impacts on the coupled system are currently under investigation and will be the subject of an other publication. To clarify the message in the manuscript we added the following in the summary/discussion:

"However, at T21 resolution, there are some local changes in precipitation, mostly located over mountainous areas. Thus, some components of the model, such as continental runoff and ultimately ocean, or vegetation, are impacted by the inclusion of the downscaling. In one simulation of 1,000 years we integrated for one particular parameter combination we obtained a modified state for the ocean and the vegetation. Though the total amount of precipitation in the northern hemisphere is not modified substantially the spatial distribution of the precipitation in the different runoff basins led to a reduction of the AMOC strength and to a shallower branch of the upper branch of the thermohaline circulation in that particular simulation. To avoid this global climate drift from the CTRL experiment, we present only 100 years of model integration ensuring a limited role of the downscaling feedbacks on the global climate. However, for longer integration, the model might need some adjusment in order to correctly reproduce the present-day state of the climate system."

Relating to the SMB computation: the downscaled climatic fields we compute in the methodology outlined in the current manuscript are being used to develop a downscaled SMB. This new SMB will take explicitly into account the sub-grid temperature and precipitation according to the local orography. With this, we aim at better reproducing the non-linear nature of the SMB and in particular the position of the ablation zone at the margin. However, due to the imprint of the coarse

resolution model into the current downscaled fields, the latter cannot be used directly into the SMB and need additional steps beyond the scope of the current study that should be seen as a first necessary step.

Figures:

Fig. 2, 5 I would have preferred if the figures showed the following difference maps: X: stands for the climate variable M: for model (M\_CTLR, M\_DOWN) O: for observational data (reanalysis) LR, HR: for low and high resolution respectively Then arrange the figure in the follow 3x2 grid: left column HR, right column LR top row: observations O middle row: M\_DOWN bottom row: M\_CTRL In addition then the corresponding difference maps in a 2x2 grid left column HR, right column LR top row: difference M\_DOWN - O bottom row: difference M\_CTRL - O

Thank you for the suggestion. We rearranged the figures as suggested in the new version of the manuscript. However, we prefer to keep our labelling on the map to keep the source information. Following Dr. Fyke (reviewer #2) suggestion, we added zoom figures of precipitation over Europe (Fig. 7) and over Greenland (Fig. 8). We think that thanks to these additional figures and given the fact that the differences between the different model outputs are large, a figure with the differences is not really needed. However, such a figure is shown below.

---

## Author Comment (AC2) · 13 Oct 2017

We warmly thank the anonymous reviewer #1 and Dr. Jeremy Fyke (reviewer #2) for their insightful comments. We did our best to follow the suggestions which greatly improved the manuscript to our opinion.

In the following, we reply point by point to each individual comment (referee's comments are italicised). Following our responses, please find our new manuscript in which we highlighted changes from the original version.

**Dr. Jeremy Fyke**

This manuscript describes a method to downscale temperature and precipitation from a coarse EMIC grid to a higher resolution grid, such as (for example) that typical for ice sheet models, but also other aspects (e.g. mountainous environments). This is a very pertinent, needed, and surprisingly difficult topic, and it's great the authors are tackling this in ILOVECLIM.

I think some work is still required before this can be published, so suggest a number of revisions. Some general themes that require improvement in my opinion, and that are reflected in my more detailed comments, are: -Given the 'EMIC' nature of the model and simplifications within the downscaling scheme, I think greater emphasis on caveats associated with EMIC-embedded downscaling should be more clearly described, so that potential users are clearly aware of what aspects of their scientific simulations are a result of model simplifications, versus real processes.

Following Referee #1' suggestions, we have included a synthetic review of EMICs abilities and limitations. We hope this will help the reader and the potential users to grasp the field of application and its limits of such models. A side note is that even if the downscaling is indeed an important problem in EMIC, and more generally in Earth System Model, the different groups provide only limited information on how they deal with it.

-A clearer high-level, but technical overview of the scheme would be really useful in the Introduction, so the readers enter the details with a pre-existing, rough, mental construct of what to expect. E.g. 'Briefly, the downscaling procedure reimplements the original ECBILT equations at 11 vertical levels, followed by: : : and: : :' Also, repeat this overview at the end.

Thank you for the suggestion. We have now added these sentences in the introduction:

"The methodology chosen for the downscaling procedure is to first replicate the original model physics on artificial surfaces of a vertically extended grid. Then from the vertically extended grid, we compute the precipitation explicitly taken into account the sub-grid orography following the original model physics."

And in the conclusion:

"The methodology chosen for the downscaling procedure is to replicate the relevant parts of the model physics needed for the temperature and precipitation on the high resolution grid."

-Several important equations are presented without justification (e.g. 'we used this formulation because: : :'). Even if Haarsma et al. (1997) is cited, it would still be useful to provide a physical reasoning for the equation form.

Most of the equations have now been updated with a physical justification. Thanks for noting this caveat.

-Given the advertised capability of the downscaling, to enable science to be done at sub grid scales (e.g. GrIS ablation zones: : :) it seems necessary to show some 'zoomed-in' plots of particular regions, for example, GrIS. As it stands, the reader has to squint to try and see the downscaled fields, as they are represented across the whole domain.

The suggested maps have been added for the precipitation over Greenland and over Europe.

-Some evaluation concerns: â`A` T>there appears there is a non-monotonic saw-tooth or wave pattern in precipitation across Greenland, generated by the precipitation downscaling. This would be very problematic for actual SMB fields used for science. These features, and other potentially similar 'native grid artifact' features over non-ice-sheet regions (which I am less familiar with) need to be explained in more detail.

We added a discussion on this native grid artefact when we first present the maps of simulated

temperatures (p. 9 l. 16-20):

"In many locations, the native grid is still visible on the NH40 model results. This is because our downscaling mostly redistribute the temperature within a coarse grid point according to the subgrid elevation starting from the coarse grid information. This generates discontinuities when moving from two neighbouring cells. Only air advection, which tends to be larger along parallels than meridians, reduces the imprint of the coarse grid."

This is indeed equivalent to the phenomenon you describe in the figure you kindly shared with us (mentioned in one of your later comment, p. 18). To our knowledge the only way to get rid off these artificial features is to perform some kind of interpolation on the coarse grid values prior any downscaling. However this approach is not suitable for conservative schemes since the interpolation would not be mass conservative with respect to humidity.

Regarding the SMB, this is indeed an issue as the fields presented in the manuscript are not suitable for an ice sheet model forcing. We are currently working on the coupling methodology and hopefully will come up with an hybrid solution that ensures mass conservation and smoothed SMB fields.

P2L20: Importantly, I don't think the method is downscaling heat. It is downscaling temperature - which is not the same as heat.

It is fair to say that most of the terms in the energy balance are not downscaled in our methodology. Latent and sensible heat fluxes in *i*LOVECLIM are largely computed from temperature, but they also depend on winds, which are not downscaled. In the revised version of the manuscript we do no longer argue that we downscale heat.

P2L25: "closed water budget" -> "closed global water budget"

Done.

P2L29: The scheme could also be important for dynamic vegetation models (i.e. by resolving sub gridded elevationdependent vegetation distribution envelopes)

Yes, we now explicitly mention vegetation dynamics here.

P3L23: Review the justification for basing the linear temperature profile on the log of pressure (even if already stated in Haarsma et al (1997) and/or basic knowledge)

We added the following:

"Assuming hydrostatic equilibrium and using the ideal gas law and hydrostatic equilibrium, the temperature varies linearly with the log of pressure. For this reason, from the 650 hPa and ..."

P4L1: why is the interpolated T500 used to obtain the near-surface temperature, instead of T650 (which is presumably closer to the surface)?

Using T500 or T650 will not change the computation at the surface because in any case we assume a linear profile passing exactly through T650. However, the altitude of the 500 hPa level is set constant and homogeneous in the model, whilst the altitude of the 650 hPa is not computed explicitly it is thus easier to compute from T500 while not changing the result.

P4L1: Explain why different treatments are applied to near-surface temperature, versus T(p). Does this result in a discontinuity in temperatures, when comparing near-surface temperatures to surface temperatures? Perhaps I'm just confused here.

Eq. 3 directly derives from Eq. 1 and 2, only expressed in term of altitude of the pressure level instead of pressure. As such, there are no discontinuities. We slightly modified the text to make this clearer:

" As in Haarsma et al. (1997), the near-surface air temperature is computed from T500, using Eq. 1

to eliminate the pressure variable in the hydrostatic equilibrium equation:"

P4L9: "Due to orography, the atmospheric isotherms are shifted upwards": is there a citation explaining this physical phenomenon? Given it's role in motivating implementation of the f\_s factor, it seems important to explain.

We agree that this sentence was confusing. The main physical reasoning behind this parameter is that the along-slope temperature lapse rate is generally smaller than the free-atmosphere temperature lapse rate (e.g. Marshall et al. 2007 or Gardner et al. 2009 from observational networks in the Arctic; Minder et al. 2010 for the Cascade Mountains). In ECBilt, the near-surface air temperatures for the virtual surfaces are retrieved using the free-atmosphere temperature lapse rate (Eq. 2 in the manuscript). The use of this lapse rate instead of the along-slope one lead to an overestimation of the range of sub-grid temperatures inside a coarse grid cell. In the model, the parameter  $f_s$  is specifically designed to reduce the value of the atmospheric temperature lapse rate.

In the revised manuscript, we do no longer mention the isotherms but we explain better the reasoning of this f\_s parameter:

"The vertical lapse rate in temperature computed in the model in Eq. 2 is representative of the freeatmosphere temperature variations. Since the along-slope lapse rate is generally smaller than the free-atmosphere lapse rate (e.g. Marshall et al. 2007; Gardner et al. 2009; Minder et al. 2010), its use lead to an overestimation of the temperature changes with elevation. In order to artificially reduce the value of the vertical lapse rate in the model, we apply a global tunable correcting factor,  $f_s$  in Eq. 4 (typically ranging from 0.5 to 1.), to the orography on the vertically extended grid."

Equations 3 and 4: nearly duplicative. Could the authors just present Equation 4, then say "f\_s=1, reduces to the original equation of Haarsma et al. (1997)"?

Eq. 3 and Eq. 4 are indeed nearly duplicative except that the altitude is in one case the elevation on the native grid  $(\bar{z}_h)$  and in the other it is the altitude of the vertically extended grid  $(z_h(l=1,11))$ . We prefer to keep the two equations for sake of clarity, but we slightly modified the text around these sentences:

"In Haarsma et al. (1997) the near-surface air temperature of an atmospheric grid cell,  $\overline{T}_*$ , is computed from T500, using Eq. 1 to eliminate the pressure variable in the hydrostatic equilibrium equation:"

"With  $\overline{z}_h$  is the grid-cell surface height and z500 the height of the 500 hPa levels (prescribed homogeneously at 5500 m)"

And later:

"This equation is used to assess the near-surface air temperature for the 11 artificial surfaces using explicitly their altitude,  $z_h$ (l=1,11), instead of the actual surface height of the grid cell:"

Equations 3 and 4: why is T\* over-barred in Eq 3, but not in Eq 4?

This is to make a distinction between the fields (temperature and altitude) on the native grid (overbarred) and on the vertical levels (regular). Again, the the slight text modification presented in the previous comment hopefully facilitate the reading.

P4L13: For completeness, perhaps explicitly state these energy balance terms, indicating which ones use as input the downscaled near-surface air temperature.

**We added this information as:**

"From this near-surface air temperature for the artificial surfaces, we derive several surface energy balance terms (downward longwave radiation, latent and sensible heat flux) in the same way as Haarsma et al. (1997)".

P4L19: "In ECBilt,: : :" -> "In the idealized ECBilt representation of the atmosphere,: : :"

Done.

P4L20: What are the actual z\_h heights? I don't actually see them specified anywhere in the manuscript...

We have added this information in the text at the beginning of Sect. 2.2: "This grid consists in 11 vertical levels at 10, 250, 500, 750, 1000, 1250, 1500, 2000, 3000, 4000 and 5000 m."

Equation 5: could g be taken out of the integral (also, in the original Haarsma equation)?

**We have done so as *g* is indeed constant in the model.**

General: suggest including web link for Haarsma et al. (1997) in reference, as it is not available by, e.g. DOI. As it is it takes some brief Google searching to find a PDF copy.

**Done.**

General: how does scheme work for elevations greater than 5500m (500 hPa)? Since mountainous regions are of specific interest, and many of these regions have elevations greater than 5500m, this would seem important to note.

It depends on which grid. On the native grid, elevations greater than 500hPa are not allowed and any points above have to be cut-off. However, at T21 model resolution this situation is hardly reached; only when coupled to the ice sheet model with extreme ice sheet scenarios. On the 40 km grid, there is no computational limitation. However, we impose the value at the height of the last level in the vertically extended grid to any sub-grid point with an elevation greater than that level.

Equation 6: it's not immediately clear why the authors need to calculate surface pressure as a function of surface temperature? Can't the authors apply a more direct pressure/elevation relationship? If I'm wrong (quite possibly) - perhaps a clearer description of why this equation form is used, could be useful.

To our knowledge, there is no direct formulation p=f(z) that relates the pressure to the elevation. Any correspondence between these two variables require an assumption on the temperature as p=f(z,T). Even in simple formulation, such as the barometric formula, an hypothesis on the vertical profile of temperature has to be done. In our manuscript, Eq. 6 is simply Eq. 1 but expressed in term of pressure instead of temperature, so the use of this form is entirely consistent with the general formulation in the model. We slightly modified the text to clarify the link with Eq. 1: "The surface pressure p\_0(l=1,11) is computed rearranging Eq. 1 in term of pressure and using Eq. 2:"

General: the switching between use of values at the 650 hPA and 500 hPa levels is somewhat confusing. Can motivation/clarification be given on why this switching occurs?

Done.

Equation 7: is s\_a(k) 'surface' or 'surface area'?

We replaced surface by 'surface area'.

Equation 7: I'm not sure the initial 1/k\_max term is correct here.

Thank you for spotting it ; mistake is corrected in the revised version.

Equation 7: the authors could just say 'the surface elevation of the native grid comprises the area-weighted average of all k sub-grid points.', could they not?

Yes, we added this before the equation.

Equation 8: I'm not sure it's necessary to write out the equation for linear interpolation here. I'd be OK with just saying 'linearly interpolate from the bounding vertical levels' or something similar.

We followed your suggestion and remove this equation from the text.

General: some devil's advocate points that might be worth addressing: why not just compute T\*, Ts, and q\_max at each sub grid point, using the equations described earlier? What is the advantage of first calculating these at specified levels, then vertically interpolating? Related: others (e.g. in Fyke et al. (2011) didn't use 11 globally constant levels, but rather calculated the range of levels at each point, based on the high resolution topography within each native grid cell. This significantly reduced computation #s (for example, over large flat regions) and also allowed for finer vertical resolution in the sub gridded levels. Was this approach considered here?

The computation at each sub-grid point is a possibility. However, with the 40 km grid we have generally more than 100 sub-grid points for one native grid cell. In doing so, we will increase the computation time by a factor of ten, which is not negligible.

About the methodology, we went for the fixed levels approach mimicking the original ECBilt formulation that has fixed artificial levels for the radiative scheme. We acknowledge the fact that the adaptive vertical grid strategy of Fyke et al. is certainly better in principle. We consider this approach for further development of the downscaling methodology. However, one should keep in mind that the profiles computed in iLOVECLIM are relatively linear and thus a finer vertical resolution will not change drastically the behaviour of the model.

P5L17: how computationally expensive would it be, really, to bin sub grid points by aspect (in relation to wind direction), especially in the context of the full coupled model cost?

A first development version were we considered this possibility at an earlier stage would have required to bin sub-grid points by aspect related to wind direction at every atmospheric model time step. Since the quantified cost of the call to the sorting represent slightly more than 0.02% of the cost of one atmospheric year, this initial version would have represented something like 50% of the atmosphere. We considered this at the time to be excessive and did not go down that road. Now thinking about it, a reasonable approximation might be to pre-compute a certain number of defined, wind directions related, bin aspects and project the wind at each atmospheric time step to the closest one. Using a 8 to 10 wind directions would increase the computational cost of the atmosphere by roughly 0.5%. This is certainly doable and we are considering it, though it will take some time to implement correctly for any sub-grid provided. We feel it is thus beyond the reach of the present reviewing process but thank the reviewer of the stimulating thoughts that have arisen from his review. As a conclusion, the implementation of such a parametrisation of winds is planned, at least for testing its importance on the final fields.

P5L20: so, for leeward slopes, this means that the lowest leeward slopes receive more than high leeward slopes (and leeward slopes get the same precip as same-elevation windward slopes). These should be explicitly discussed, as a representative 'caveat' of the scheme.

This is now discussed below.

P5L20: also, in my understanding, it is 'mid-level' windward slopes that often get the most precipitation. Whereas the described scheme bias precipitation to low levels. Another caveat to emphasize.

In fact, the lowest elevation grid points have more precipitable water but their saturation is more difficult to reach because of higher temperature. It is thus not straightforward that our methodology bias precipitation towards low levels precipitation with respect to the mid-levels. It may also well depend on the context: seashore bordering mountains may tend to get more precipitation at low level than mid-altitude level in some specific context (e.g. Norway). Also, given the simplicity of the scheme (relying on large scale saturation), certainly do not depict all the processes that would be needed for the discrimination of precipitation of low to mid levels. One example is the fact that the model has no diurnal cycle and that for low to mid levels of the atmosphere this has a very strong impact for the actual vertical structure in temperature. We agree nonetheless that more discussion is needed and therefore added the following in the manuscript:

"As we compute precipitation for a sorted sub-grid point, we remove available precipitable water from the amount of total precipitable water of the previous grid point. In doing so, we assume that the mountain edges (lowest elevations) are the first affected by moisture influx. However, in our approach two points at the same altitude will have the same amount of precipitation, independently from the wind direction. The model is thus intrinsically unable to reproduce high precipitation on windward slopes and conversely low precipitation on leeward slopes. A foreseen model development will be to sort the sub-grid points depending on wind direction."

Section 2.3.2: 'Dynamic precipitation' is an unknown term to me (a non-meteorologist). If it isn't a common meteorological term, perhaps re-name, or explicitly define.

Dynamic precipitation is also known as stratiform precipitation. We now use this terminology in the manuscript.

P5L26: if the local topography exists above 500 hPA, does it receive any precipitation at all in the model?

As explained in a previous comment, in the T21 grid the topography can not go beyond 500 hPa. In the fine grid, a point with a topography higher than 500 hPa can receive precipitation, computed from the higher vertically extended point.

P5L27: "expended"->"expanded"

Replaced by "extended" for consistency with the rest of the manuscript.

Equation 12: it's not clear what the physical justification is for the form of this equation. Please describe in greater detail, with citations.

This parameter is introduced in order to artificially mimic the elevation desertification (less precipitation due to lower moisture content), an aspect commonly observed over large orographic features such as ice-sheets. This notion is hardly represented in *i*LOVECLIM because of the intrinsic model assumptions. Given that in the model all the moisture exceeding the saturation is used to form precipitation and that saturation is lower at high altitude because of lower temperature, the model tends to produce higher precipitation over elevated region. For large orographic features, the air masses tend to become drier from the edges towards the interior. However, because of the coarse resolution of the model, there is still a large moisture advection within the large orographic features. As such, this parameter is introduced to somehow tune the modelled precipitation, changing the distribution between low lying areas (which present a dry bias generally) and mountains (wet bias).

General: which of the 3 contributors to precipitation (2x dynamic, and convective) is generally the dominant term? This would be useful for readers to know.

Convective precipitation represents roughly 10% of the total precipitation. This is now explicitly mentioned in the manuscript:

"Convective precipitation is assumed to be an adjustment term to reach stability in the atmospheric column. They represent roughly 10% of the total precipitation in the model."

P7L2: It's not clear to me how the convective precipitation scheme works. For example, given the repeated use of Eq. 13, where does the assessment of stability come in? I think a clearer description is needed here.

We largely changed the text in the description of the convective scheme with clarity in mind. The new version reads:

"We compute convective precipitation after the stratiform precipitation. If the moisture availability  $q_a(k=1,k_{max})$  is still greater than  $\alpha_q(k)$   $q_{max}(k)$  then the amount of convective precipitation,  $p_{conv}(k=1,k_{max})$ , is computed with the same formulation as in Eq. 12. As for the stratiform precipitation, the convective precipitation is associated with a local heat release affecting the temperature at 350 hPa,  $T_{350}(k=1,k_{max})$ . After this convective precipitation, we assess stability comparing the moist adiabatic lapse rate to the local potential temperature at 500~hPa,  $\theta(k=1,k_{max})$ , computed from the potential temperatures at 350 hPa and 650 hPa. The stability is assessed for each individual sub-grid points. If the stability is not reached, we allow a new convective precipitation term computed from  $q_a(k=1,k_{max})$ . The heat release in the upper atmosphere at each

precipitation event tends to increase stability. This is an iterative process and we only go to the next sub-grid point when we reach stability locally."

P8L2: by 40%, for the whole coupled model? Or just the atmospheric component?

For the whole (standard) coupled model: atmosphere, ocean, vegetation (no carbon cycle nor ice sheet models). This is clarified in the text.

Section 3.1.1/3.2: note the caveat that the authors are comparing a preindustrial simulation to recent historical climatologies (or describe why this isn't a caveat, e.g., why recent historical climatology is close enough to preindustrial climatology for the fields in question, to warrant direct comparison)

We agree than it would have been ideally better to use modern simulations in order to have atmospheric fields more directly comparable to observations. However, the modern climate is far from equilibrium and in such modern simulations it would have been more difficult to isolate the sole effect of the downscaling compared to the combined effects of the different forcing acting together. In addition, the model biases are generally much larger than the temperature and precipitation differences from a modern and pre-industrial simulations.

P7L16: given 'continentality' is usually associated with sub-annual ranges in temperatures, what does it mean to interpret 'continentality' over Siberia, when using annual mean fields? Furthermore, it is unclear how this relates to other regions (as it is written, it seems to indicate that increased Siberian continentality causes biases elsewhere)?

We acknowledge the improper use of continentality here. We changed it for: "Whilst the model reproduces the cold temperatures in Siberia, it is elsewhere generally largely too warm, in particular over North America, Greenland and Western Europe."

P7L18: "does not imply important changes in surface temperature": relative to the default CTRL case? Perhaps reword for clarity.

Rephrased to:

"On the other hand, at the continental scale, our downscaling procedure does not imply important changes in surface temperature relative to the CTRL experiment.."

P7L30: Given the 'ijk' indexing is hardly used in the manuscript (as figures mostly show the results only from one ensemble), I'm wondering how useful it is to describe this indexing scheme? It would become more useful, if plots of results as a function of parameter space, were shown: : :

A plot of the metrics (correlation, standard deviation and root mean square errors) in the parameter space has been now added in the manuscript. Also, following your suggestion, we removed the ijk notation, and mention the parameter values explicitly.

Figure 2/4/5/6: why was DOWN020 chosen as the representative plot? How much do the different ensemble members look, and why/why not?

DOWN020 was chosen because it produces a good compromise in terms of the various metrics. We now hope that the plot you suggested (metrics in the parameter space) gives more information on the different ensemble members look like and why our choice is a good compromise.

Figure 2, 'DOWN NH40' panel: it is surprising to see remnants of the native grid in many places, though I suspect I know why. I think a description of why this remaining imprinting occurs should be clearly explained to readers.

We hope that we have answered your comment when replying to your previous concern on this native grid artefact (p. 1-2 on this document). Also, we have added the following discussion when describing this figures (p. 9 I. 16-20):

"[However, the downscaling induces important local temperature changes, particularly visible on the NH40 grid.] In many locations, the native grid is still visible on the NH40 model results. This is because our downscaling mostly redistribute the temperature of a coarse grid point according to the sub-grid elevation starting from the coarse grid information. This generates discontinuities when moving from two neighbouring cells. Only air advection, which tends to be larger along parallels than meridians, reduces the imprint of the coarse grid."

Figure 3: The green/blue shading is quite confusing to parse, visually. Could shaded 'clouds' be more visually accessible?

We apologize but we did not understand what "shaded clouds" are in this comment. Therefore we kept the current color shading which presents the advantage of consistency and allows distinguishing between  $f_s=0.6$  (green) and  $f_s=1.0$  (blue).

P8L26: ": : : : to correct the model bias: : : " -> ": : : to correct broader region model biases that are unrelated to topographic forcing: : : "

Done, thank you for the suggestion.

P8L27: ": : :horizontal gradients: : :"->": : :horizontal temperature gradients: : :"

Done.

General: given the advertised ability of the scheme to downscale high-resolution mapview T/P fields, and the intention of the authors that the downscaling will improve 'regional' studies, it would be useful to see regional subset plots of Figure 2, ERA-interim and DOWN NH40. For example, given my background, I'd like a closer look at GrIS precipitation!

We have followed your suggestion and added regional maps of precipitation for Greenland and Western Europe.

Figure 4: A better description of the Taylor diagram scheme would be good. For example, is 'standard deviation' the standard deviation in mean annual values, for the long-term climatology (or is it describing the standard deviation in temporal variability?). Similarly for the correlation axis.

We added the following information in the caption of the two Taylor Diagram:

"In this figure, the metrics (standard deviation, correlation and root mean square error) are computed from the annual mean climatic variables. The standard deviation in the observations is used to normalise the standard deviations and the root mean square error."

P9L8: describe why the lack of impact on native-grid model performance is a good/bad thing.

The original model has been generally tuned on various variables (not only T and P). A drastic change in the model response could require to perform the global tuning again. This is explicitly mentioned at the end of Sect. 3.

General: a more robust description of the analysis of the full 50-member analysis is warranted. For example, which varied parameter makes the most difference? Is it possible to identify parameter combinations that are optimal, for particular locations?

We now discuss the impact of the different parameters in more details through the description of the figure showing the metrics in the parameter space. All parameters influence the results while the  $f_s$  parameter may be seen as slightly more important than the others. It is indeed possible to find a set of parameters that work better for a specific region of interest but might provide poorer score in different region. In particular in regions with a dry bias in the model, a set of parameters that produce more precipitation will generally also produce more precipitation in regions with a wet bias. As stated in the manuscript, this is because the downscaling have only a limited impact on the large scale behaviour of the model.

We added the following:

"The downscaling performance with respect to CRU~CL-v2 is also shown in Fig. 11 in which we present quantitative metrics (spatial correlation, standard deviation and root mean square error) as a function of parameter values. The parameters that have the strongest influence on the simulated

precipitation are  $f_s$  and  $\alpha_q^{min}$ . A lower value for these parameters tend to produce higher spatial correlation, lower standard deviation and lower root mean square error. However, for  $z_q$ =2000m, low values for the two other parameters can lead to an underestimation of the standard deviation. The standard deviation and the root mean square error have a similar response to a change in parameters, whilst the spatial correlation is mostly sensitive to the  $\alpha_q^{min}$  parameter, with higher correlation for lower value of this parameter."

Figure 5: contrary to the text, it looks to me like DOWN NH40 \*does\* better resolve the Norwegian/W. North America high-precipitation bands: : :

We produce indeed more precipitation over the mountains in these areas but we fail at reproducing the strong increased just at the coast. The text in this section has been largely modified.

General: A stronger justification is needed that downscaling does indeed produce 'scientifically useful' high-resolution precipitation fields. As it is, the reader is somewhat left to their own devices to piece together the various impacts of precipitation downscaling into a coherent story on how well the precipitation downscaling scheme (perhaps the most important but tenuous aspect of the whole procedure) performs, and whether it would help/hinder their scientific simulations.

As stated in the introduction of the manuscript, this downscaling has been initially developed for a better coupling between the atmosphere and the ice sheet model. The downscaled climatic fields we compute in the methodology outlined in the current manuscript are being used to develop a downscaled SMB. This new SMB will take explicitly into account the sub-grid temperature and precipitation according to the local orography. With this, we aim at better reproducing the non-linear nature of the SMB and in particular the position of the ablation zone at the margin. However, as you also mention, due to the imprint of the coarse resolution model into the current downscaled fields, the latter cannot be used directly into the SMB and need additional steps beyond the scope of the current study that should be seen as a first necessary step. We believe also that the present study is "scientifically useful" in that it reproduces better the large spatial variability over marked orographic features such as the Alps.

We added in the conclusion, first paragraph:

"We have shown that the inclusion of the downscaling allows for a better representation of the precipitation, especially for mountainous region."

And in the last paragraph of the discussion:

"[From the downscaled atmospheric fields, we are now able to compute the surface mass balance required by the ice sheet model embbeded in *i*LOVECLIM]. This downscaled surface mass balance explicitly take into account the sub-grid temperature and precipitation according to the local orography. With this, we aim at better reproducing the non-linear nature of the SMB and in particular the position of the ablation zone at the margin. Foreseen applications include ice sheet - climate interactively coupled thanks to the downscaled atmospheric fields. "

Figure 6: The GrIS cross section highlights some concerning 'sawtooth/wave' behavior, whereby the downscaled precipitation field changes across the boundaries of the native grid. To be honest, we have struggled with a similar (?) thing in CESM, and I ended up putting together this Google-based schematic that tries to explain our particular problem: https://docs.google.com/presentation/d/1gyalZ5ypZ3XWxf2VTThuBmkpL9Qf18Qg5w qpbzPm6s/edit#slide=id.p Non-monotonic SMB fields that could certainly overwhelm the positive impacts of downscaling over GrIS, and preclude use of the downscaling scheme for science, where the GrIS-wide SMB field is important. Please comment on why the 'sawtooth/wave' pattern occurs in the downscaling, and why the authors consider it acceptable for subsequent science using downscaled SMB fields.

Thank you for kindly sharing this figure with us. This is exactly the problem we are facing. We have addressed your concern on p. 1-2 in this document when describing Fig.3 (temperature maps) as these features also arise for the temperature field.

Figure 7: as with figure 4: a better description and interpretation of the Taylor diagram would seem important. Also, it's quite hard to pick out salient differences and understand them in a broader context, given the selection of what seems like an arbitrary subsample the whole ensemble.

We have provided more information on the construction of normalised Taylor diagrams. Also, we think that the figure you suggested (metrics in the parameter space) provide now the broader context to interpret the model response.

Figure 7: it's not clear now the 'spatial correlation is greatly improved', via inspection of the Taylor diagrams. Which dots should the reader compare, to see this impact? Perhaps this is just my eternal personal struggle with Taylor diagrams, though :).

To improve the correlation in the Taylor diagram you have to move along a circle, clockwise. We add the values in the text, for the reader to spot where to look at:

"The spatial correlation is in particular generally greatly improved (from about 0.25 to more than 0.4)."

Figure 4: why not use the same set of ensemble members, as in Figure 7 (for consistency, and perhaps just to show that temperature downscaling is not as parametrically sensitive as precipitation downscaling)?

This was our first version of this figure. However, the dots were all almost superposed and it was impossible to distinguish between them. We prefer to keep the figure this way for clarity of the figure.

P10L20: Yes, it is notable how downscaling significantly impacts precipitation on the native grid (e.g. CTRL T21 vs. DOWN T21). For example, it appears North America as a whole receives quite a bit more precipitation when downscaling is utilized. Yet this important aspect is not clearly mechanistically explained. A physical reasoning behind this non-negligible impact should be described in the manuscript.

On the one hand, a large part of North America present a rough topography. Because of the exponential shape of the saturation function as a function of temperature, we can expect greater precipitation even if the T21 topography is unchanged. On the other hand, we also modified the  $\alpha_q^{min}$  parameter which controls how much saturated we should be to initiate precipitation for low land points. The CTRL version of the model has a similar parameter, which is set to 0.9. To illustrate the effect of this parameter, we show below (Fig. A3), a similar figure than Fig. 6 but with  $\alpha_q^{min} = 0.9$ .

---

## Author Response (AR2)

We answer J. Fyke additional comments below.

*-Abstract: I don't think it's correct in this case to state the scheme conserves energy and moisture - particularly the former. Conservation of energy entails much more than conservative interpolation of temperature. For example, can iLOVECLIM, that includes the downscaling scheme, demonstrate a globally closed energy budget, including, e.g., latent heat transformations (such that the net energy change in the system is equal to the net TOA flux)?*

The remark of the reviewer point to the fact, we think, that we did not explain clearly enough the energy conservation we claimed in the downscaling scheme and, associated to this, in the atmospheric model. Nowhere in the manuscript did we pretend that we have an energy conservation "through the interpolation of temperature" as suggested by the reviewer. Indeed, we are not interpolating temperature in an horizontal direction, but only in a vertical direction prior to energy conservation calculations. If the reviewer takes the meaning of energy conservation as identical energy fluxes before and after the downscaling, this is clearly not the case. However, in the meaning we imply "conservation of energy through the different components, though distributed differently", which is exactly the case.

In ECBilt, by design, the energy is conserved since the heat flux towards land and ocean is computed from the difference of the incoming radiations (shortwave and longwave) and the outcoming radiave (longwave) and turbulent (latent and sensible) heat fluxes. This ensures a strict energy conservation. The downscaling typically change the repartition of energy between the different components (particularly modifying the latent heat flux component), but does not modify the conservation scheme. Hence the claim that the downscaling scheme does not modify the energy conservation is correct.

We simply suggest this in the abstract:

"Our scheme is non grid-specific and conserves energy and moisture **in the same way as the original climate model**".

We hence modified the text of the manuscript with clarification in mind as follow: (p.3 l. 4-5):

"Computed on each atmospheric timestep, the downscaling accounts for the feedback of sub-grid precipitation on large scale energy and water budget. **Whilst the energy repartition between the turbulent fluxes is modified, the conservation is ensured however in the same way as in ECBilt, where the heat flux towards land and ocean is computed as the imbalance between the incoming (both shortwave and longwave) and the outgoing radiation (longwave only) as well as the turbulent (latent and sensible) heat fluxes. The conservation of energy and water is [...]**"

*-Abstract (and elsewhere): In the Reply to Reviewers the authors write (note paraphrasing) "Regarding the SMB, [grid imprinting issues] presented in the manuscript are not suitable for an ice sheet model forcing" and "Due to the imprint of the coarse resolution model into the current downscaled fields, the latter cannot be used directly into the SMB and need additional steps beyond the scope of this current study…" Yet in the public-facing manuscript, the authors repeatedly state: "Foreseen applications of this new model feature includes ice sheet model coupling…". There is an apparent disconnect here, between what the authors honestly think their model is capable of, and how its capabilities are described in the manuscript. The authors need to be more explicit with readers about the applicability of their developments, \*as they stand\* - for example, if the work presented here is considered an interim benchmark on the way to something that is scientifically useful for, e.g., ice sheet SMB science, this needs to be clearly stated.*

We added the following text to the conclusion, again with clarity in mind:

"Foreseen applications include ice sheet - climate interactively coupled thanks to the downscaled atmospheric fields **although the artificial discontinuities due to the imprint of the coarse native grid cell in the downscaled field are still an important drawback of the method presented**. Ice sheet mass balance is not the only [...]"

*-On grid imprinting: I think the reasons for actual grid imprinting are still not described clearly enough for the general reader. Regarding temperature, for example: "This is because our downscaling mostly redistribute (sic) the temperature of a coarse grid point according to the sub-grid elevation starting from the coarse grid information" - is an ambiguous sentence that leaves the reader puzzled (and does not imply that this problem also extends to precipitation, as noted in the Reply to Reviews). Regarding precipitation: even if you took winds into account, it would still be true that the main (only?) effect of downscaling is to redistribute precipitation within a native T21 grid cell. Thus, T21 grid cell imprinting will almost certainly still occur across the location of T21 grid cell boundaries in downscaled precipitation.*

*As above, I feel the authors should be more transparent with describing the caveats of their approach, particularly with respect to remaining grid imprinting - just so that potential users aren't surprised/frustrated when actually applying the scheme (or run into issues when they submit science using the scheme to peer review).*

We totally agree: the addition of wind direction in our scheme would not suppress the imprint of the native grid. We simply stated in the manuscript that the inclusion of wind direction could improve the precipitation distribution within a given coarse grid pixel.

We also want to point out that even if the main effect of the downscaling is to redistribute the precipitation within a native T21 grid cell according to the sub-grid topography, we think that this is a considerable improvement from the base model.

We rephrased the  sentence the reviewer is pointing to as follow:

"

**The imprint of the native grid remains because the primary effect of the downscaling is to physically compute the distribution of the climatic variables linked to temperature and precipitation according to the sub-grid topography for a given coarse grid information. By design, this generates discontinuities when moving from two neighbouring cells.** "

We added at the end of the paragraph (p. 9 l. 20): "This imprint is also present in the precipitation field (Sec. 3.2.2)."

*-There remain a fairly large number of grammatical errors that the authors could sweep for prior to any official GMD proof-read.*

A native English speaker has now checked the revised version of the manuscript.

[revised manuscript text omitted]